# FedReLa: Imbalanced Federated Learning via Re-Labeling

## Abstract

Federated learning has emerged as the foremost approach for decentralized model training with privacy preserving. The global class imbalance and cross-client data heterogeneity naturally coexist, and the mismatch between local and global imbalances exacerbates the performance degradation of the aggregated model. The agnosticism of global minority classes poses significant challenges for data-level methods, especially under extreme conditions with severe class deficiencies across clients. In this paper, we propose FedReLa, a novel data-level approach that tackles the coexistence of data heterogeneity and class imbalance in federated learning. By re-labeling samples with a feature-dependent label re-allocator, FedReLa corrects the biased decision boundaries without requiring knowledge of the global class distribution. This modular, model-agnostic approach can be integrated with algorithmic methods to offer consistent improvements without any extra communication burden. Through extensive experiments, our method significantly improves the accuracy of minority classes and the overall accuracy on step-wise-imbalanced and long-tailed datasets, outperforming the previous state of the art.

## 1 Introduction

Federated learning (FL) facilitates collaborative model training across distributed clients without exchanging raw data, thereby preserving data privacy. Each client trains models locally on private data and uploads only parameter updates to a global server. However, due to variations in client environments such as differences in the received data, participation capacity, and geographic or demographic differences, among others, local data often exhibit significant heterogeneity, leading to disparate parameter updates and suboptimal global model convergence (Zhao et al., 2018).

Imbalanced data, where some classes have many samples while others have few, is common in real-world applications (Azaria et al., 2014; Fotouhi et al., 2019; Shingi, 2020) and even more prevalent in FL. Due to client-level data heterogeneity, two types of imbalance often coexist: local imbalance (within individual clients) and global imbalance (across the entire federation), both of which pose challenges for FL classification. Prior research has primarily addressed local imbalance through improved aggregation (McMahan et al., 2017; Wang et al., 2020), robust local training (Acar et al., 2021; Karimireddy et al., 2019; Li et al., 2021; 2020), selective client participation (Chen et al., 2020; Fraboni et al., 2021), or architectural enhancements (Duan et al., 2019). However, these approaches typically assume all the classes are equally represented, a condition rarely met in practice.

Recent works focus on more realistic scenarios where global class imbalance (e.g., step-wise or long-tailed distributions) *coexists* with data heterogeneity. Early works such as Ratio-loss (Wang et al., 2021) and CLIMB (Shen et al., 2021) pioneered solutions for step-wise global imbalance under non-IID (not independent and identically distributed) client data. Subsequent studies (Chen & Chao, 2021b; Li et al., 2023; Shang et al., 2022; Xiao et al., 2024; 2023) further tackled federated long-tailed (Fed-LT) learning. While these works acknowledge both global class imbalance and heterogeneity, they primarily focus on algorithm-level classifier enhancements (e.g., tailored aggregation rules or client-specific optimization). *Existing data-level methods typically rely on class prior information.* For instance, feature-level SMOTE techniques (Chawla et al., 2002b) require prior knowledge of which classes are majority or minority to synthesize new samples. *In FL, however, such information is often unavailable due to privacy constraints, especially in the presence of global-local imbalance mismatches.* Although in some domain-specific scenarios (e.g., fraud detection (Shingi, 2020) or

rare disease diagnosis (Tan et al., 2023)), the minority class can be identified due to its natural rarity, server data heterogeneity often results in the absence of minority classes in many clients. Such class absence makes it impossible to synthesize minority class samples that do not exist locally. *Therefore, data-level approaches without access to global distribution statistics remain underdeveloped.*

This paper addresses one of the most prevalent yet challenging scenarios in FL: improving FL under the coexistence of agnostic *data heterogeneity* and *mismatched global-local class imbalance*, *without additional communication cost and extra local training burden caused by additional optimizable model parameters*. We propose a novel data-level approach that re-labels local data through a carefully designed feature-dependent label re-allocator. Specifically, our label reallocating mechanism re-labels the majority class samples that intrude into the global minority-class feature spaces, thereby implicitly enlarging the minority-class decision boundary. The key innovation lies in the design of our label re-allocator, which leverages the knowledge of the minority class from the global model to asymmetrically re-label majority class samples based on their posterior probabilities estimated from local data. Unlike traditional data-level augmentation methods, such as SMOTE-based (Chawla et al., 2002b; He et al., 2008) or mixup-based approaches (Chou et al., 2020; Ramasubramanian et al., 2024), our method operates purely in the label space, without synthesizing new features.

We present FedReLa, a model-agnostic approach for addressing heterogeneous, class-imbalanced data in **Fed**erated Learning via **Re-La**beling, with three defining characteristics:

(i) **Model-Data Agnosticism**: Unlike methods that rely on balanced auxiliary data or explicit class priors (Shingi, 2020; Wang et al., 2021), FedReLa is agnostic to model architecture, data format, and class distribution and inherently improves data quality without domain-specific constraints.

(ii) **Nearly-Zero-Cost Plug-in Adaptation**: Unlike prior methods (Duan et al., 2019; Shang et al., 2022; Shen et al., 2021; Xiao et al., 2024) that introduce additional communicational or training cost from optimizing and uploading newly introduced parameters or modules, FedReLa re-purposes the global model as a label re-allocator without introducing additional trainable parameters, requiring no extra training or communicational burden. The only computational cost of FedReLa is the one-time model inference required to obtain posterior probabilities, which can be naturally collected during training epochs. Such a one-shot computation is negligible compared to the whole training process (see Appendix B.1). Furthermore, FedReLa efficiently balances the global classifier in the fine-tune stage without retraining the model, while all computation and re-labeling are done locally in parallel.

(iii) **Universal Composability**: Operating solely in the label space, FedReLa integrates seamlessly with algorithm-level approaches and delivers consistent performance gains.

We evaluate the performance of FedReLa through extensive experiments on Fashion-MNIST, CIFAR-10, and CIFAR-100 under both step-wise and long-tailed class distributions, across varying degrees of data heterogeneity and imbalance ratios. FedReLa consistently enhances prior algorithm-level methods, achieving state-of-the-art performance with negligible additional computation cost, while avoiding extra communication or parameter training overhead. Notably, in the most extreme cases, FedReLa boosts minority/tail-class accuracy by up to 38.30% (step-wise) and 30.7% (long-tailed) while maintaining overall accuracy superiority (shown in Tables 1 and 2 in Section 4). These results conclusively demonstrate the superiority and applicability of FedReLa in practical FL deployments.

**Related works.** *(1) Centralized imbalance learning on decentralized data.* Imbalance learning has seen significant success in centralized settings. The model-agnostic advantages of data-level methods are utilized in data preprocessing to augment features of minority class samples by either generating new samples via generative models (Odena et al., 2017; Mariani et al., 2018) and SMOTE-based methods (Chawla et al., 2002a; Han et al., 2005; He et al., 2008) or mixing existing sample features by mixup-based approaches (Chou et al., 2020; Ramasubramanian et al., 2024; Zhang et al., 2017). However, in heterogeneous decentralized data scenarios, the effectiveness of augmenting local data of these methods becomes very limited. Data heterogeneity, limited seed samples, and class absence on local clients severely restrict their ability to generate minority class samples. Without access to global class priors, ReMix (Chou et al., 2020) fails to identify global minority classes for proper adjustment of label mixup strength, while SelMix (Ramasubramanian et al., 2024) faces practical constraints due to its reliance on auxiliary balanced validation data, which in many cases is scarce. Similarly, algorithm-level methods like loss reweighting (Tan et al., 2020) or logit adjustments (Li et al., 2022) also fall short in FL due to the lack of access to the global label distribution.

*(2) Federated learning with data heterogeneity.* Three pivotal components in federated learning frameworks critically influence global model performance: client update, model aggregation, and local datasets. Numerous FL methods have been developed to address local data heterogeneity and imbalance, primarily focusing on client update and model aggregation to mitigate the adverse effects of skewed local datasets. FedAvg (McMahan et al., 2017) and FedNova (Wang et al., 2020) pioneered weighted averaging during model aggregation based on local dataset sizes or batch sizes. In the paradigm of modifying client update, regularization terms are incorporated into loss functions to penalize discrepancies between global and local models (Li et al., 2021; 2020) or constrain inter-round model divergence (Acar et al., 2021). SCAFFOLD (Karimireddy et al., 2019) introduced control variates to correct biased local gradients. However, these methods exhibit suboptimal performance on global minority/tail classes, as they primarily address local imbalance induced by data heterogeneity while neglecting global class imbalance.

*(3) Data heterogeneity with global class imbalance.* Several approaches have adapted loss reweighting strategies to address global imbalance. Ratio-Loss (Wang et al., 2021) estimates global class priors using an auxiliary balanced dataset. To eliminate the reliance on auxiliary data, CLIMB (Shen et al., 2021) optimizes the local model with additional learnable loss-weighting parameters, but increases the local training workload and communication cost. Recent studies extend this problem to long-tailed distributions. CReFF (Shang et al., 2022) enhances tail-class performance by retraining classifiers with aggregated class-specific features, consequently introducing additional local training overhead and doubled communication costs. FedROD (Chen & Chao, 2021b) further extends the scope to personalized FL by decoupling the training of the global and personalized models by separately optimizing the local models with balanced softmax and cross-entropy loss. However, blindly fully balancing the local loss may lead to a suboptimal global model. To improve the performance of both global and personalized models on long-tailed data, FedETF (Li et al., 2023) replaces the classifier head with a fixed ETF (Equiangular Tight Frame) to enforce the learning of balanced features.

Based on the observation that head classes tend to have larger weight norms, FedGraB (Xiao et al., 2023) rescales the gradients of local models by class weight norms to enhance tail-class performance. As a follow-up improvement of (Xiao et al., 2023), FedLOGE (Xiao et al., 2024) further integrates the idea of (Li et al., 2023) by rescaling the weights of the fixed ETF classifier using the weight norms of an auxiliary classifier head. Despite this, our empirical findings in Section 4 reveal that weight norms become unreliable under high heterogeneity.

**Motivations.** Existing methods tackle data imbalance through algorithmic adjustments. Why not improve local data quality directly? The reason is apparent: conventional data augmentation relies on global data distribution knowledge, which violates FL privacy constraints. The most relevant work, FedMix (Wicaksana et al., 2022), addresses heterogeneity using mixup. Still, it requires clients to share local data averages, which may require additional privacy-preserving mechanisms and increase the communication costs. To our knowledge, no existing data-level FL approach mitigates the coexistence of data heterogeneity and global imbalance while achieving: (1) not requiring auxiliary datasets, (2) having a negligible additional computation cost with zero communication cost and no extra parameter training burden, and finally (3) being agnostic to global data distribution. This motivates FedReLa, a data-level method that simultaneously achieves all of the above requirements while significantly improving performance under extreme conditions.

## 2 Federated Learning via Re-Labeling: FedReLa

In this section, we first analyze how local and global imbalances affect decision boundaries and why heterogeneity in globally imbalanced data exacerbates the performance impact of imbalance data on global models. We then introduce the label re-allocator and analyze how re-labeled samples implicitly rebalance the biased global decision boundaries.

### 2.1 Problem Formulation

Consider a dataset $\mathcal{D}$ that contains data pairs $(X, Y) \sim P(x, y)$, where $x \in \mathcal{X} \subseteq \mathbb{R}^d$, $y \in \mathcal{Y} = \{1, 2, \ldots, C\}$ and $P$ represents the joint distribution. Denote the conditional distribution $X \mid Y = j \sim P_j(x)$ and the prior probability $\Pr(Y = j) = \pi_j$ for class $j \in \mathcal{Y}$. The marginal distribution of $X$ is then $P_X(x) = \sum_{j \in \mathcal{Y}} \pi_j P_j(x)$. Assume $\mathcal{D}$ is imbalanced with global imbalance

ratio $\text{IR}(\mathcal{D}) = \max_{j \in \mathcal{Y}} \pi_j / \min_{j \in \mathcal{Y}} \pi_j \gg 1$. Let $\eta_j(x) = \Pr(Y = j \mid X = x) = \pi_j P_j(x)/P_X(x)$ be the global posterior probability. Recalling that the Bayesian decision theorem (Duda et al., 2006) defines the optimal estimated $y^*$ of a sample $x$ as $y^* = \arg\max_{j \in \mathcal{Y}} \eta_j(x)$, the following result holds.

**Lemma 1.** *The optimal Bayesian decision boundary between two classes $j \neq \ell \in \mathcal{Y}$ is*

$$S_{j,\ell} = \{x \in \mathcal{X} : \eta_j(x) = \eta_\ell(x) > \eta_{\ell'}(x) \ \forall \ \ell' \in \mathcal{Y} \setminus \{j, \ell\}\}.$$

For $x \in S_{j,\ell}$, it holds that $\eta_j(x) = \eta_\ell(x)$ implying $P_j(x)/P_\ell(x) = \pi_\ell/\pi_j$. Then for some minority class $j$ and majority class $\ell$ with $\pi_j \ll \pi_\ell$, $S_{j,\ell}$ intrudes deeply into the minority class region, increasing the risk of misclassifying minority class samples. This motivates balancing the ratio $\pi_\ell/\pi_j$ to *shift the decision boundary back towards the majority class region*, thereby alleviating the adverse effects of class imbalance.

In FL, the dataset $\mathcal{D}$ is distributed on $K$ clients with local datasets $\{\mathcal{D}^{(k)}\}_{k=1}^K$ and assumes the class conditional distributions $\{P_j^{(k)}(x)\}_{k=1}^K$ are identical across all clients for each $j \in \mathcal{Y}$. In contrast, the class priors $\{\pi_j^{(k)}\}_{k=1}^K$ may be different among clients due to data heterogeneity. For two classes $j$ and $\ell$, we have $P_j^{(1)}(x)/P_\ell^{(1)}(x) = \cdots = P_j^{(K)}(x)/P_\ell^{(K)}(x) = P_j(x)/P_\ell(x)$. However, divergent class priors result in different local posterior probability $\eta_j^{(k)}(x) = \pi_j^{(k)} P_j(x)/\sum_{j_0 \in \mathcal{Y}} \pi_{j_0}^{(k)} P_{j_0}(x)$ and misaligned Bayesian decision boundaries among clients. This misalignment affects the performance of the aggregated classifier and slows down the convergence rate of FL algorithms (Zhao et al., 2018).

Ideally, with properly chosen aggregation weights $\{w_k\}_{k=1}^K$, the decision boundary between classes $j$ and $\ell$ of the global aggregated model $\eta_j^{[w]}(x) = \sum_{k=1}^K w_k \eta_j^{(k)}(x)$ given by

$$S_{j,\ell}^{[w]} = \{x \in \mathcal{X} : P_j(x)/P_\ell(x) = \pi_\ell^{[w]}/\pi_j^{[w]}\},$$

can match the decision boundary $S_{j,\ell}$ in Lemma 1 by making $\pi_j^{[w]} = \pi_j$ and $\pi_\ell^{[w]} = \pi_\ell$, where $\pi_j^{[w]} = \sum_{k=1}^K w_k \pi_j^{(k)} / \sum_{k=1}^K w_k$ and $\pi_\ell^{[w]} = \sum_{k=1}^K w_k \pi_\ell^{(k)} / \sum_{k=1}^K w_k$. For instance, setting $w_k = |\mathcal{D}^{(k)}|/|\mathcal{D}|$ achieves this alignment. However, the global imbalance ratio $\pi_\ell/\pi_j$ still introduces bias into the aggregated decision boundary of the global model. To address that, several algorithms (Menon et al., 2020; Tan et al., 2020) have been proposed to adjust the ratio $\pi_\ell^{[w]}/\pi_j^{[w]}$ via alternative weighting schemes. Moreover, data heterogeneity can cause mismatches between global and local imbalance ratios, further complicating the class imbalance issue and amplifying bias in the aggregated decision boundary. See Example 1 in the Appendix for an illustration.

## 2.2 AGGREGATED DECISION BOUNDARY WITH RE-LABELED DATA

In this paper, we introduce a novel data-level approach that reallocates data labels to adjust the decision boundary by balancing class prior ratios at both local and global levels. This strategy also alleviates the mismatch between global and local imbalance ratios, and improves the overall robustness of the FL model. Our proposed FedReLa is motivated by how re-labeling shifts decision boundaries locally and globally. We begin by analyzing its effect on a local client $k$, and then extend the discussion to model aggregation. Without loss of generality, we consider a binary classification setting where classes $j$ and $\ell$ represent the minority and majority classes, respectively.

Let $(X^{(k)}, Y^{(k)}) \sim P^{(k)}(x, y)$ denote the data pair for client $k$ with re-labeled $\widetilde{Y}^{(k)}$ and consider $\widetilde{\mathcal{D}}^{(k)}$ the corresponding re-labeled dataset. Denote the probabilities of re-labeling $\ell$ to $j$ as $\rho_{\ell \to j}^{(k)}(x) = \Pr(\widetilde{Y}^{(k)} = j \mid X^{(k)} = x, Y^{(k)} = \ell)$ and re-labeling $j$ to $\ell$ as $\rho_{j \to \ell}^{(k)}(x) = \Pr(\widetilde{Y}^{(k)} = \ell \mid X^{(k)} = x, Y^{(k)} = j)$ for client $k$, then

$$\widetilde{\eta}_j^{(k)}(x) = \Pr(\widetilde{Y}^{(k)} = j \mid X^{(k)} = x) = \eta_j^{(k)}(x)[1 - \rho_{j \to \ell}^{(k)}(x)] + [1 - \eta_j^{(k)}(x)]\rho_{\ell \to j}^{(k)}(x).$$

**Lemma 2.** *The optimal Bayesian decision boundary based on $\widetilde{\mathcal{D}}^{(k)}$ for client $k$ is*

$$\widetilde{S}^{(k)} = \left\{ x^* \in \mathcal{X} : \frac{P_j(x^*)}{P_\ell(x^*)} = \frac{1 - 2\rho_{\ell \to j}^{(k)}(x^*)}{1 - 2\rho_{j \to \ell}^{(k)}(x^*)} \cdot \frac{\pi_\ell^{(k)}}{\pi_j^{(k)}} \right\},$$

*provided that $\rho_{\ell \to j}^{(k)}(x^*) \leq 0.5$ and $\rho_{j \to \ell}^{(k)}(x^*) \leq 0.5$ for any $x^* \in \widetilde{S}^{(k)}$.*

When $\pi_\ell^{(k)}/\pi_j^{(k)} \gg 1$, we seek to achieve $[1 - 2\rho_{\ell \to j}^{(k)}(x^*)]/[1 - 2\rho_{j \to \ell}^{(k)}(x^*)] < 1$ to locally adjust the decision boundary. Given the scarcity of minority class samples in $\mathcal{D}^{(k)}$, it is reasonable to restrict re-labeling to occur only from majority class $\ell$ to minority class $j$, and set $\rho_{j \to \ell}^{(k)}(x) = 0$. Furthermore, since deeply invaded majority class samples are especially harmful, we design a label re-allocator where the re-labeling probability is proportional to the degree of this intrusion. Specifically, with $\rho_{\ell \to j}^{(k)}(x) \propto \eta_j^{(k)}(x)$, the Bayesian decision boundary becomes

$$\widetilde{S}^{(k)} = \left\{ x^* \in \mathcal{X} : P_j(x^*)/P_\ell(x^*) = [1 - 2\rho_{\ell \to j}^{(k)}(x^*)]\pi_\ell^{(k)}/\pi_j^{(k)} \right\}.$$

Since $1 - 2\rho_{\ell \to j}^{(k)}(x^*) < 1$, the boundary $\widetilde{S}^{(k)}$ on re-labeled data, shifts back to the majority class region. Based on re-labeled data $\widetilde{\mathcal{D}} = \cup_{k=1}^{K} \widetilde{\mathcal{D}}^{(k)}$, we also study the decision boundary of the global aggregated model $\widetilde{\eta}_j^{[w]}(x) = \sum_{k=1}^{K} w_k \widetilde{\eta}_j^{(k)}(x)$.

**Lemma 3.** *The optimal Bayesian decision boundary of the global aggregated model $\widetilde{\eta}_j^{[w]}(x)$ is*

$$\widetilde{S}^{[w]} = \left\{ x^* \in \mathcal{X} : \frac{P_j(x^*)}{P_\ell(x^*)} = \frac{\sum_{k=1}^{K} w_k \pi_\ell^{(k)}[1 - 2\rho_{\ell \to j}^{(k)}(x)]/\pi_\ell}{\sum_{k=1}^{K} w_k \pi_j^{(k)}/\pi_j} \cdot \frac{\pi_\ell}{\pi_j} \right\}.$$

Lemma 3 implies that the label re-allocator balances the global imbalance ratio when

$$\frac{\sum_{k=1}^{K} w_k \pi_\ell^{(k)}[1 - 2\rho_{\ell \to j}^{(k)}(x)]/\pi_\ell}{\sum_{k=1}^{K} w_k \pi_j^{(k)}/\pi_j} < 1. \tag{1}$$

By choosing $w_k = |\mathcal{D}^{(k)}|/|\mathcal{D}|$, we have $\sum_{k=1}^{K} w_k \pi_j^{(k)}/\pi_j = 1$, and (1) reduces to $\sum_{k=1}^{K} w_k \pi_\ell^{(k)}[1 - 2\rho_{\ell \to j}^{(k)}(x)]/\pi_\ell < 1$, which holds naturally when $\rho_{\ell \to j}^{(k)}(x) > 0$ for all $k \in \{1, \dots, K\}$. Thus, the label re-allocator can balance both the local and global decision boundary.

**Remark 1.** *Due to data heterogeneity, local class distribution can deviate significantly from the global one. It is possible for a class that is globally a minority to become a majority within certain clients. As local clients lack access to the global class prior ratios, the mismatch can lead to re-labeling in unexpected directions. For instance, when re-labeling class $j$ samples to class $\ell$ even if $\pi_\ell \gg \pi_j$ globally. To handle this, we let the re-labeling direction be determined by local priors: on client $k$, if $\pi_\ell^{(k)} > \pi_j^{(k)}$, then class $\ell$ samples are re-labeled to class $j$, and vice versa. This results in the following Bayesian decision boundary on client $k$:*

$$\widetilde{S}^{(k)} = \left\{ x^* \in \mathcal{X} : \frac{P_j(x^*)}{P_\ell(x^*)} = \frac{1 - 2\rho_{\ell \to j}^{(k)}(x^*) \cdot \mathbb{I}(\pi_\ell^{(k)} > \pi_j^{(k)})}{1 - 2\rho_{j \to \ell}^{(k)}(x^*) \cdot \mathbb{I}(\pi_\ell^{(k)} < \pi_j^{(k)})} \cdot \frac{\pi_\ell^{(k)}}{\pi_j^{(k)}} \right\}. \tag{2}$$

*Then, the Bayesian decision boundary of the global aggregated model $\widetilde{\eta}_j^{[w]}(x)$ takes the form:*

$$\widetilde{S}^{[w]} = \left\{ x^* \in \mathcal{X} : \frac{P_j(x^*)}{P_\ell(x^*)} = \frac{\sum_{k=1}^{K} w_k \pi_\ell^{(k)}[1 - 2\rho_{\ell \to j}^{(k)}(x)]\mathbb{I}(\pi_\ell^{(k)} > \pi_j^{(k)})/\pi_\ell}{\sum_{k=1}^{K} w_k \pi_j^{(k)}[1 - 2\rho_{j \to \ell}^{(k)}(x)]\mathbb{I}(\pi_\ell^{(k)} < \pi_j^{(k)})/\pi_j} \cdot \frac{\pi_\ell}{\pi_j} \right\}.$$

*Under global imbalance where $\pi_\ell \gg \pi_j$, we typically observe that $\sum_{k=1}^{K} \mathbb{I}(\pi_\ell^{(k)} > \pi_j^{(k)}) > \sum_{k=1}^{K} \mathbb{I}(\pi_\ell^{(k)} < \pi_j^{(k)})$, meaning more clients locally reflect the global imbalance than contradict it. Furthermore, even if $\pi_\ell^{(k_0)} < \pi_j^{(k_0)}$ for some client $k_0$, its weight $w_{k_0} \propto |\mathcal{D}^{(k_0)}|$ is often small as $|\mathcal{D}^{(k_0)}|$ is less than double of the total number of class-$j$ samples in the full dataset $\mathcal{D}$. As a result, we still expect a correction in the decision boundary of the global aggregated model with $\frac{\sum_{k=1}^{K} w_k \pi_\ell^{(k)}[1 - 2\rho_{\ell \to j}^{(k)}(x)]\mathbb{I}(\pi_\ell^{(k)} > \pi_j^{(k)})/\pi_\ell}{\sum_{k=1}^{K} w_k \pi_j^{(k)}[1 - 2\rho_{j \to \ell}^{(k)}(x)]\mathbb{I}(\pi_\ell^{(k)} < \pi_j^{(k)})/\pi_j} < 1.$*

## 3 FRAMEWORK OF FEDRELA

Motivated by decision boundary adjustment through data re-labeling, as analyzed in Section 2.2, we propose FedReLa to mitigate performance degradation caused by data heterogeneity and class

imbalance in FL. FedReLa is an adaptive and model-agnostic approach, which is designed as a plug-in module that can be seamlessly integrated into any existing FL algorithm.

As shown in Figure 1, FedReLa works as a local data one-shot preprocessor between communication rounds of any FL algorithm, with each client applying it locally and in parallel. Specifically, before client $k$ starts to train the global model $f(\theta; x)$ with parameter $\theta = \theta_t^{\text{global}}$ received at round $t = T_{\text{relabel}}$, FedReLa re-labels its local dataset $\mathcal{D}^{(k)} = \{(x_i^{(k)}, y_i^{(k)})\}_{i=1}^{n_k}$ using a client-specific label re-allocator $\rho^{(k)}$, resulting in the re-labeled dataset $\widetilde{\mathcal{D}}^{(k)} = \{(x_i^{(k)}, \widetilde{y}_i^{(k)})\}_{i=1}^{n_k} = \rho^{(k)}(\mathcal{D}^{(k)})$. Note that the computation of FedReLa only occurs in $T_{\text{relabel}}$, and the re-labeled local dataset $\widetilde{\mathcal{D}}^{(k)}$ can be reused in subsequent training rounds $t > T_{\text{relabel}}$. Thus, the one-shot computations at round $T_{\text{relabel}}$ for each client-specific label re-allocator are lightweight and almost negligible to the whole training process. We discuss the approximate computational cost of FedReLa in Appendix B.1. Each client then updates the global model locally using $\widetilde{\mathcal{D}}^{(k)}$, and the server aggregates the local updates $\Delta\theta_t^{(k)}$ to produce the updated global model with parameter $\theta_{t+1}^{\text{global}}$.

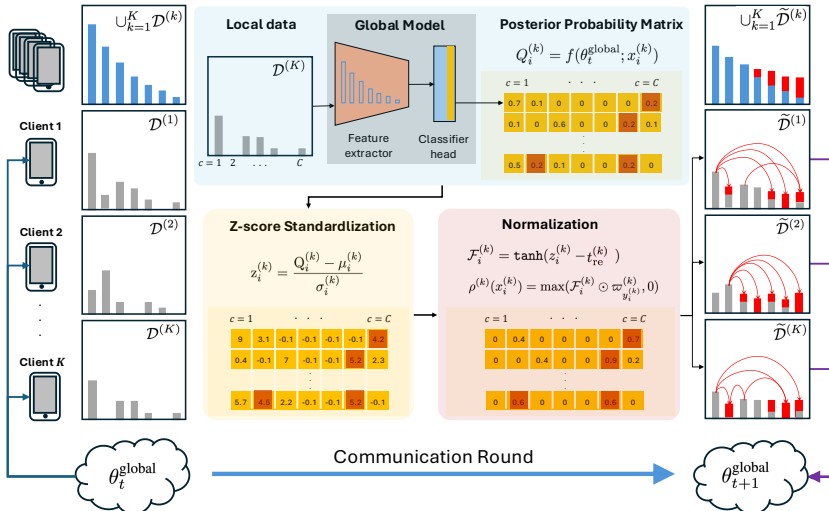

Figure 1: FedReLa Framework. At round $t = T_{\text{relabel}}$, FedReLa re-labels the local dataset with the label re-allocator based on the global model before the local training starts.

The inspiration of FedReLa is to "reallocate" the shared feature space that is encroached upon by the majority class (due to biased decision boundaries) to the minority class. This is achieved by selectively re-labeling the majority-class samples that intrude into the minority-class feature space with similar features as minority-class labels. Building upon the theoretical analysis in Section 2, we let the re-labeling probabilities be proportional to the posterior probabilities of minority classes, and we utilize the global model distributed to each client to perform local inference. This yields a $|\mathcal{D}^{(k)}| \times |\mathcal{Y}|$ posterior probability matrix $\mathbf{Q}^{(k)}$ for client $k$, where

$$Q_i^{(k)} = f(\theta_t^{\text{global}}; x_i^{(k)}) \in \mathbb{R}^{|\mathcal{Y}|}$$

is the $i$-th row of $\mathbf{Q}^{(k)}$, denoting the posterior probability vector of the $i$-th local instance $x_i^{(k)}$.

Crucially, the global model implicitly integrates cross-client discriminative knowledge across all classes $\mathcal{Y} = \{1, 2, \ldots, C\}$, making it assign non-zero posterior probabilities even to classes absent from a client's local data. Due to global imbalance and data heterogeneity, these posterior estimates for minority (tail) classes tend to be systematically biased downward. To address posterior underestimation and obtain a well-calibrated label re-allocator, we introduce two key normalization steps: (1) z-score Standardization, and (2) `tanh` Normalization.

**Class-wise z-score Standardization.** Samples near decision boundaries often share features with other classes, thus exhibiting relatively high posterior probabilities for ambiguous ones. As a result of biased global decision boundaries towards minority classes, most dominant-class samples exhibit

---

**Algorithm 1** Local training process for client $k$ with FedReLa at communication round $t$

---

**Input:** local epochs $E$, learning rate $\eta$, local datasets $\mathcal{D}^{(k)} = \{(x_i^{(k)}, y_i^{(k)})\}_{i=1}^{n_k}$, classifier $f(\cdot; \cdot)$, global model parameter $\theta_t^{\text{global}}$

**Parameters:** Threshold $t_{\text{re}}^{(k)}$, re-labeling round $T_{\text{relabel}}$

---

*Client $k \in K$ executes:*

$\texttt{ClientUpdate}(t, \mathcal{D}^{(k)}, \theta_t^{\text{global}})$:
**if** $t == T_{relabel}$ **then**
$\quad | \quad \widetilde{\mathcal{D}}^{(k)} \leftarrow \texttt{ReAllocator}(\mathcal{D}^{(k)}, \theta_t^{\text{global}})$
$\quad | \quad \mathcal{D}^{(k)} \leftarrow \widetilde{\mathcal{D}}^{(k)}$
**end**
$\theta_t^{(k)} \leftarrow \theta_t^{\text{global}}$
**for** *epoch $e = 1 \rightarrow E$* **do**
$\quad |$ **for** *each batch $b \in \mathcal{D}^{(k)}$* **do**
$\quad | \quad | \quad \theta_t^{(k)} \leftarrow \theta_t^{(k)} - \eta \nabla \mathcal{L}(\theta_t^{(k)}; b)$
$\quad |$ **end**
**end**
$\Delta \theta_t^{(k)} \leftarrow \theta_t^{\text{global}} - \theta_t^{(k)}$
**return** $\Delta \theta_t^{(k)}$

$\texttt{ReAllocator}(\mathcal{D}^{(k)}, \theta_t^{\text{global}})$:

**for** $(x_i^{(k)}, y_i^{(k)}) \in \widetilde{\mathcal{D}}^{(k)}$ **do**
$\quad | \quad Q_i^{(k)} \leftarrow f(\theta_t^{\text{global}}; x_i^{(k)})$
**end**
Compute $\varpi^{(k)}$ by (4) with $n_{\mathcal{Y}}^{(k)}$
**for** $(x_i^{(k)}, y_i^{(k)}) \in \mathcal{D}^{(k)}$ **do**
$\quad |$ Compute $z_i^{(k)}$ by (3)
$\quad | \quad \varpi_{y_i^{(k)}}^{(k)} \leftarrow \max(\varpi^{(k)} - \varpi^{(k)}[y_i^{(k)}], 0)$
$\quad |$ Compute $\rho^{(k)}(x_i^{(k)})$ by (5)
$\quad | \quad \mathcal{U} \in \mathbb{R}^{|\mathcal{Y}|} \leftarrow \texttt{Bernoulli}(\rho^{(k)}(x_i^{(k)}))$
$\quad |$ **if** $\mathcal{U}$ *contains 1* **then**
$\quad | \quad | \quad \widetilde{y}_i^{(k)} \leftarrow \mathcal{Y}[\texttt{argmax}(\rho^{(k)}(x_i^{(k)}))]$
$\quad |$ **end**
**end**
**return** $\widetilde{\mathcal{D}}^{(k)} \leftarrow \{(x_i^{(k)}, \widetilde{y}_i^{(k)})\}_{i=1}^{n_k}$

---

vanishingly small posterior probabilities for minority classes. Despite this, we empirically observe that a non-trivial subset of majority-class samples retains non-negligible probabilities for minority classes–insufficient to trigger misclassification but indicative of proximity to minority-class regions in the eature space. To better calibrate these underestimated posteriors, particularly for minority classes, we apply class-wise z-score standardization, which rescales the posterior distributions within each class. This highlights candidate samples with shared features for re-labeling. Specifically, the z-score vector for the $i$-th instance in client $k$ is computed as:

$$z_i^{(k)} = \frac{Q_i^{(k)} - \mu_i^{(k)}}{\sigma_i^{(k)}}, \ \mu_i^{(k)} = \frac{1}{|\mathcal{I}_i^{(k)}|} \sum_{i_0 \in \mathcal{I}_i^{(k)}} Q_{i_0}^{(k)}, \ \sigma_i^{(k)} = \sqrt{\frac{1}{|\mathcal{I}_i^{(k)}| - 1} \sum_{i_0 \in \mathcal{I}_i^{(k)}} \left(Q_{i_0}^{(k)} - \mu_i^{(k)}\right)^2}, \quad (3)$$

where $\mathcal{I}_i^{(k)} = \{i_0 \in \{1, \ldots, n_k\} : y_{i_0}^{(k)} = y_i^{(k)}\}$ denotes the index set of samples in $\mathcal{D}^{(k)}$ that share the same label as the $i$-th instance. Here, $\mu_i^{(k)} \in \mathbb{R}^{|\mathcal{Y}|}$ and $\sigma_i^{(k)} \in \mathbb{R}^{|\mathcal{Y}|}$ are the class-wise mean and standard deviation vectors of the posterior probabilities over this set. As illustrated in Figure 1, the resulting z-score matrix $\mathbf{Z}^{(k)}$, with $i$-th row $z_i^{(k)}$, recalibrates $\mathbf{Q}^{(k)}$, amplifying underestimated posterior probabilities of minority (tail) classes and highlighting samples near class boundaries.

**Normalization via `tanh`.** To ensure that the re-labeling rates in the label re-allocator lie within $[0, 1]$, we rescale the z-scores to the range $[-1, 1]$ using a `tanh` transformation. This normalization incorporates two critical components: (1) a client-specific threshold $t_{\text{re}}^{(k)}$, which is a tunable hyperparameter that determines the desired re-labeling strength by filtering out samples with weak feature similarity; and (2) a class-wise reweighting vector $\varpi_j^{(k)} \in \mathbb{R}^{|\mathcal{Y}|}$, computed from local class priors to re-label samples asymmetrically. Specifically, $\varpi_j^{(k)}$ reweighs the re-labeling probability from class $j$ to any other class $c \in \mathcal{Y} \setminus j$ with their class prior difference between $j$ and $c$.

Let $n_{\mathcal{Y}}^{(k)} \in \mathbb{R}^{|\mathcal{Y}|}$ denote the vector of class-wise sample counts in the local dataset $\mathcal{D}^{(k)}$. We first apply min-max normalization on $n_{\mathcal{Y}}^{(k)}$ to construct the class-wise reweighting vector $\varpi_j^{(k)}$, defined as:

$$\varpi_j^{(k)} = \max(\varpi^{(k)} - \varpi^{(k)}[j], 0) \ \text{with} \ \varpi^{(k)} = 1 - \texttt{minmax}(n_{\mathcal{Y}}^{(k)}). \quad (4)$$

For samples belonging to a local tail class $y_{\text{tail}}$, we have $\varpi^{(k)}[y_{\text{tail}}] = 1$, which implies that $\varpi^{(k)}_{y_{\text{tail}}} = 0$. It zeros the re-labeling probabilities from the minority class to other classes, thereby preserving the integrity of minority-class samples.

With a client-specific $t^{(k)}_{\text{re}}$, we define the re-labeling probability for the $i$-th instant in client $k$ as

$$\rho^{(k)}(x_i^{(k)}) = \max\left(\tanh(z_i^{(k)} - t^{(k)}_{\text{re}}) \odot \varpi^{(k)}_{y_i^{(k)}}, 0\right), \tag{5}$$

where $\odot$ denotes element-wise multiplication, and $t^{(k)}_{\text{re}}$ acts as a tunable filtering threshold to suppress re-labeling for samples with low similarity to the minority-class feature. Notably, when the z-score falls below $t^{(k)}_{\text{re}}$, the re-labeling probability becomes negative and is truncated to zero. We performed a sensitive analysis on $t^{(k)}_{\text{re}}$ in Appendix B.3.

Based on the local label re-allocator $\rho^{(k)}$, each client applies probabilistic re-labeling to its data to generate the re-labeled data before local training; this is the only difference FedReLa makes from the standard FL, which adjusts the decision boundaries and thus achieves significant improvement on the performance of minority/tail classes. The full procedure is summarized in Algorithm 1.

## 4 EXPERIMENTS

**Datasets.** To provide a comprehensive evaluation, we conduct experiments under both **step-wise** and **long-tailed** global imbalance settings on Fashion-MNIST (F-MNIST) Xiao et al. (2017), CIFAR-10 Krizhevsky & Hinton (2009), and CIFAR-100 Krizhevsky & Hinton (2009) datasets. For step-wise imbalance, we undersample 10% or 30% of the classes with an imbalance ratio (IR) of 10 or 20. For long-tailed imbalance, the datasets are sampled into a long-tailed class distribution using an imbalance factor (IF) of 50 or 100 as in (Cao et al., 2019). To simulate cross-client heterogeneity, we employ latent Dirichlet sampling (Chen & Chao, 2021a) to partition the data in a non-IID fashion across clients. Specifically, we use $K = 100$ clients for the step-wise versions of Fashion-MNIST and CIFAR-10, and $K = 40$ for their long-tailed versions. For CIFAR-100, we use $K = 10$ clients in both step-wise and long-tailed settings. The heterogeneity level is controlled by the parameter $\alpha \in \{0.1, 0.3, 10\}$. We set the client sample rate to 1. (See evaluation metrics in Appendix B.)

**Baseline and prior SOTA.** We compare FedReLa with prior baselines and SOTA methods under both step-wise and long-tailed imbalance settings. For step-wise imbalance, we evaluate against FedAvg (McMahan et al., 2017), FedProx (Li et al., 2020), FedNova (Wang et al., 2020), MOON (Li et al., 2021), and CLIMB (Shen et al., 2021). For long-tailed imbalance, we compare with FedETF (Li et al., 2023) and the latest SOTA method, FedLOGE (Xiao et al., 2024). As FedReLa handles data heterogeneity and class imbalance in FL at the data level, our method can seamlessly integrate with the above methods, offering further improvements. We thus compare methods trained on original-labeled data with those trained on re-labeled data by FedReLa. All methods are trained with sufficient communication rounds to converge. Please refer to Appendix B for the communication rounds needed to achieve the convergence of each method.

**Performance comparision.** For step-wise imbalance scenarios, Table 1 shows that FedReLa consistently enhances accuracy for both minority classes and overall performance across varying imbalance ratios (IR) and minority class proportions at heterogeneity level of $\alpha = 0.3$ (see Section C in the Appendix for ablation analysis on $\alpha$). On Fashion-MNIST and CIFAR-10, FedReLa achieves 6.40%–32.20% minority-class accuracy improvement and 0.81%–4.76% overall accuracy gain under IR $= 10$. At IR $= 20$, the approach further elevates minority-class accuracy by 11.83%–38.30% and overall accuracy by 1.46%–7.79%. For CIFAR-100, FedReLa delivers a steady 6.03%–15.04% boost in minority-class accuracy while maintaining overall accuracy superiority. We also notice that the majority-class accuracy experiences some degradation in the 30%-minority-class setting, as the strong performance of minority classes inherently compromises inflated majority-class performance. This aligns with the fundamental trade-off in class-imbalance learning: enhancing minority-class performance necessarily diminishes the over-privileged majority-class performance, a characteristic shared by all imbalance methods. For the 10%-minority-class scenario, FedReLa exhibits a negligible impact on majority-class accuracy and even improves it on CIFAR-10. This stems

from label rectification by FedReLa, which relieves class overlap and thereby reduces outlier-induced interference for majority classes, particularly on clients with local-global IR mismatch.

| Dataset | IR | Methods | 10% Minority | | | 30% Minority | | |
|---|---|---|---|---|---|---|---|---|
| | | | Majority | Minority | Overall | Majority | Minority | Overall |
| F-MNIST | 10 | FedAvg | 88.43(87.40)-1.03 | 52.50(77.00)+24.50 | 84.84(86.36)+1.52 | 89.86(90.17)+0.31 | 60.67(70.50)+9.83 | 81.10(84.27)+3.17 |
| | | FedProx | 88.23(87.43)-0.80 | 53.20(76.80)+23.60 | 84.73(86.37)+1.64 | 90.60(88.14)-2.46 | 60.07(70.40)+10.33 | 81.44(82.82)+1.38 |
| | | FedNova | 86.81(87.46)+0.65 | 67.10(77.50)+10.40 | 84.84(86.46)+1.62 | 88.41(86.83)-1.58 | 69.77(76.17)+6.40 | 82.82(83.63)+0.81 |
| | | MOON | 88.41(87.83)-0.58 | 47.00(73.00)+26.00 | 84.27(86.35)+2.08 | 90.61(89.24)-1.37 | 59.80(69.87)+10.07 | 81.37(83.43)+2.06 |
| | | CLIMB | 89.05(89.98)+0.93 | 65.52(76.24)+10.72 | 86.70(88.61)+1.91 | 93.00(92.30)-0.70 | 67.23(75.47)+8.24 | 85.27(87.25)+1.98 |
| | 20 | FedAvg | 88.64(88.07)-0.57 | 49.00(73.10)+24.10 | 84.68(86.57)+1.89 | 90.44(87.33)-3.11 | 50.50(75.70)+25.20 | 78.46(83.84)+5.38 |
| | | FedProx | 89.37(87.77)-1.60 | 44.58(73.60)+29.02 | 84.89(86.35)+1.46 | 90.69(87.37)-3.32 | 50.00(74.90)+24.90 | 78.48(83.63)+5.15 |
| | | FedNova | 88.46(88.37)-0.09 | 52.34(71.30)+18.96 | 84.85(86.66)+1.81 | 85.94(87.27)+1.33 | 55.03(77.90)+22.87 | 76.67(84.46)+7.79 |
| | | MOON | 89.07(88.07)-1.00 | 32.40(66.60)+34.20 | 83.40(85.92)+2.52 | 91.13(88.63)-2.50 | 44.43(74.77)+30.34 | 77.12(84.47)+7.35 |
| | | CLIMB | 90.30(90.32)+0.02 | 51.28(71.34)+20.06 | 86.40(88.43)+2.03 | 94.34(90.28)-4.05 | 53.27(73.60)+20.33 | 82.02(85.28)+3.26 |
| CIFAR-10 | 10 | FedAvg | 60.10(59.84)-0.26 | 27.80(55.70)+27.90 | 56.87(59.43)+2.56 | 65.93(62.14)-3.79 | 22.67(41.33)+18.66 | 52.95(55.90)+2.95 |
| | | FedProx | 60.66(61.18)+0.52 | 30.02(58.70)+28.68 | 57.60(60.93)+3.33 | 67.66(60.79)-6.87 | 22.80(45.10)+22.30 | 54.20(56.08)+1.88 |
| | | FedNova | 58.54(58.60)+0.06 | 29.90(57.00)+27.10 | 55.68(58.44)+2.76 | 65.23(62.70)-2.53 | 23.47(39.20)+15.73 | 52.70(55.65)+2.95 |
| | | MOON | 58.62(60.33)+1.71 | 17.10(49.30)+32.20 | 54.47(59.23)+4.76 | 66.67(63.21)-3.46 | 23.63(38.67)+15.04 | 53.76(55.85)+2.09 |
| | | CLIMB | 81.62(82.68)+1.06 | 37.45(46.18)+8.73 | 77.20(79.03)+1.83 | 86.82(87.47)+0.65 | 33.59(43.26)+9.67 | 70.85(74.21)+3.36 |
| | 20 | FedAvg | 60.33(60.70)+0.37 | 17.25(51.60)+34.35 | 56.02(59.79)+3.77 | 67.39(61.51)-5.88 | 13.82(47.97)+34.15 | 51.32(57.45)+6.13 |
| | | FedProx | 59.33(60.58)+1.25 | 15.60(53.90)+38.30 | 54.96(59.91)+4.95 | 67.69(61.96)-5.73 | 15.02(47.87)+32.85 | 51.89(57.73)+5.84 |
| | | FedNova | 61.77(62.77)+1.00 | 26.05(57.80)+31.75 | 58.20(62.27)+4.07 | 66.97(60.12)-6.85 | 18.20(53.03)+34.83 | 52.34(57.99)+5.65 |
| | | MOON | 58.75(59.72)+0.97 | 10.12(47.70)+37.58 | 53.89(58.52)+4.63 | 64.51(60.77)-3.74 | 7.65(40.03)+32.38 | 47.45(54.55)+7.10 |
| | | CLIMB | 79.53(80.34)+0.81 | 28.38(40.21)+11.83 | 74.42(76.33)+1.91 | 87.75(85.81)-1.94 | 24.03(38.77)+14.74 | 68.64(71.70)+3.06 |
| CIFAR-100 | 10 | FedAvg | 58.67(58.08)-0.59 | 12.30(23.10)+10.80 | 54.03(54.58)+0.55 | 58.67(57.07)-1.60 | 14.37(25.70)+11.33 | 45.38(47.66)+2.28 |
| | | FedProx | 58.14(58.03)-0.11 | 13.00(26.00)+13.00 | 53.63(54.83)+1.20 | 58.84(58.10)-0.74 | 14.93(22.03)+7.10 | 45.67(47.28)+1.61 |
| | | FedNova | 58.67(57.90)-0.77 | 13.40(23.90)+10.50 | 54.14(54.50)+0.36 | 59.49(58.00)-1.49 | 13.53(23.23)+9.70 | 45.70(47.57)+1.87 |
| | | MOON | 57.55(57.70)+0.15 | 13.22(23.92)+10.70 | 53.12(54.32)+1.20 | 58.60(56.83)-1.77 | 16.37(23.93)+7.56 | 45.93(46.96)+1.03 |
| | | CLIMB | 47.96(48.28)+0.32 | 10.50(24.90)+14.40 | 44.21(45.94)+1.73 | 49.16(47.44)-1.72 | 10.83(25.87)+15.04 | 37.66(40.97)+3.31 |
| | 20 | FedAvg | 59.34(59.01)-0.33 | 6.80(15.80)+9.00 | 54.09(54.69)+0.60 | 58.73(57.50)-1.23 | 5.90(11.93)+6.03 | 42.88(43.83)+0.95 |
| | | FedProx | 58.86(58.18)-0.68 | 5.00(14.10)+9.10 | 53.47(53.77)+0.30 | 59.49(56.93)-2.56 | 6.03(13.23)+7.20 | 43.45(43.82)+0.37 |
| | | FedNova | 59.16(58.49)-0.67 | 7.30(17.80)+10.50 | 53.97(54.42)+0.45 | 59.60(58.11)-1.49 | 6.03(13.57)+7.54 | 43.53(44.75)+1.22 |
| | | MOON | 57.89(57.43)-0.46 | 6.65(18.02)+11.37 | 52.77(53.49)+0.72 | 59.36(56.14)-3.22 | 5.90(13.87)+7.97 | 43.32(43.46)+0.14 |
| | | CLIMB | 47.91(47.21)-0.70 | 5.02(16.25)+11.23 | 43.62(44.12)+0.50 | 49.22(46.42)-2.80 | 4.34(13.44)+9.10 | 35.76(36.53)+0.77 |

Table 1: Test accuracies (in %) in the format of `original(+FedReLa)`**+enhancement**/-`tradeoff` of different methods on step-wise imbalance datasets at heterogeneity level of $\alpha = 0.3$.

On long-tailed data, Table 2 shows that FedReLa consistently outperforms prior SOTA methods. The improvements are more pronounced under higher heterogeneity partitions, with FedReLa achieving +17.21% and +1.32% overall accuracy gains on CIFAR-10 and CIFAR-100, respectively, under the most extreme imbalanced and heterogeneous settings. Inflated majority class performance inherently comes at the expense of minority classes. Again, in most cases, the gains in overall accuracy outweigh any reductions in overstated head class accuracy, indicating a favorable trade-off.

| Dataset | IF | Heterogeneity | $\alpha = 0.1$ | | $\alpha = 0.3$ | | $\alpha = 10$ | |
|---|---|---|---|---|---|---|---|---|
| | | Method/Metrics | H/M/T-shots | Overall | H/M/T-shots | Overall | H/M/T-shots | Overall |
| CIFAR-10 | 50 | FedETF | 86.82/55.33/24.62 | 58.71 | 88.42/71.74/64.11 | 76.12 | 90.41/80.02/68.12 | 80.60 |
| | | +(FedReLa) | 75.44/69.63/61.11 | **69.47** | 84.51/76.44/69.63 | 77.64 | 89.54/79.22/71.80 | 81.14 |
| | | FedLOGE | 68.67/49.93/58.23 | 59.92 | 84.92/72.13/74.57 | 77.98 | 89.42/80.80/71.97 | 81.60 |
| | | +(FedReLa) | 64.17/64.67/73.20 | 67.03 | 78.77/81.73/78.00 | **79.43** | 87.65/80.80/77.73 | **82.62** |
| | 100 | FedETF | 33.12/63.52/20.94 | 37.33 | 92.02/69.20/54.33 | 70.14 | 93.92/74.80/59.33 | 74.32 |
| | | +(FedReLa) | 61.23/51.82/51.64 | **54.54** | 89.44/68.42/64.83 | 73.33 | 92.22/76.18/61.84 | 75.24 |
| | | FedLOGE | 36.73/38.13/42.72 | 39.55 | 89.13/70.03/64.52 | 73.56 | 92.37/74.87/67.50 | 77.17 |
| | | +(FedReLa) | 56.67/38.20/62.45 | 53.44 | 85.27/65.23/74.10 | **74.79** | 89.40/75.93/71.90 | **78.36** |
| CIFAR-100 | 50 | FedETF | 55.80/47.74/29.09 | 44.11 | 67.81/49.53/25.78 | 47.32 | 71.41/53.72/26.18 | 46.01 |
| | | +(FedReLa) | 55.34/48.12/32.33 | **45.13** | 65.24/51.02/28.55 | **47.88** | 69.14/54.63/29.35 | 47.04 |
| | | FedLOGE | 37.30/43.67/37.12 | 39.34 | 58.45/47.97/32.71 | 46.09 | 53.38/45.36/32.12 | 47.12 |
| | | +(FedReLa) | 52.91/45.79/32.32 | 43.56 | 59.45/49.41/32.51 | 46.81 | 51.56/47.45/35.224 | **47.43** |
| | 100 | FedETF | 54.70/45.11/18.1 | 36.82 | 67.22/50.79/20.30 | 42.30 | 71.82/52.21/20.60 | 42.61 |
| | | +(FedReLa) | 56.54/45.83/19.54 | **38.14** | 63.54/51.33/24.34 | **43.12** | 68.44/53.53/26.13 | **44.73** |
| | | FedLOGE | 28.03/40.37/25.85 | 30.84 | 54.89/48.03/26.07 | 40.51 | 68.67/51.46/24.70 | 43.53 |
| | | +(FedReLa) | 45.55/44.37/23.71 | 36.24 | 52.61/49.00/28.26 | 41.09 | 65.22/52.19/27.28 | 44.00 |

Table 2: Test accuracies (in %) of different methods on long-tailed CIFAR-10/100.

## 5 CONCLUSION

We propose FedReLa, a data-level approach for addressing class imbalance and data heterogeneity in FL. By asymmetrically re-labeling local data by a feature-dependent label re-allocator, FedReLa rectifies decision boundaries without relying on global class priors or additional communication. Empirical results across step-wise and long-tailed settings demonstrate consistent improvements in minority-class and overall accuracy over existing methods, especially under extreme heterogeneity. FedReLa is easy to integrate into algorithmic methods, offering a practical solution for real-world FL.

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

## A    TECHNICAL APPENDICES

| Symbol | Definition |
|---|---|
| *1. Datasets & Sets* | |
| $D$ | Global dataset (union of all local datasets) |
| $D^{(k)}$ | Local dataset of client $k$ |
| $\tilde{D}^{(k)}$ | Re-labeled local dataset of client $k$ (by FedReLa) |
| $X$ | Feature space ($x \in X \subseteq \mathbb{R}^d$) |
| $Y$ | Label space ($Y = \{1, 2, ..., C\}$, $C$: number of classes) |
| $Y^{(k)}$ | Original label set of client $k$ |
| $\tilde{Y}^{(k)}$ | Re-labeled label set of client $k$ |
| $I_i^{(k)}$ | Index set of samples in $D^{(k)}$ with the same label as $x_i^{(k)}$ |
| *2. Model & Parameters* | |
| $\theta$ | Model parameter vector |
| $\theta_t^{\text{global}}$ | Global model parameter at communication round $t$ |
| $\theta^{(k)}$ | Local model parameter of client $k$ |
| $f(\theta; x)$ | Global model (maps feature $x$ to posterior probabilities) |
| $T_{\text{relabel}}$ | Communication round for FedReLa's one-shot re-labeling |
| *3. Probability & Distribution* | |
| $\pi_j$ | Global prior probability of class $j$ ($\pi_j = \Pr(Y = j)$) |
| $\pi_j^{(k)}$ | Local prior probability of class $j$ on client $k$ |
| $\pi_j^{[w]}$ | Weighted aggregated prior of class $j$ (server-side) |
| $\eta_j(x)$ | Global posterior probability of class $j$ given $x$ |
| $\eta_j^{(k)}(x)$ | Local posterior probability of class $j$ given $x$ on client $k$ |
| $\tilde{\eta}_j^{(k)}(x)$ | Posterior probability of class $j$ on $\tilde{D}^{(k)}$ |
| $\tilde{\eta}_j^{[w]}(x)$ | Aggregated posterior probability of class $j$ (server-side) |
| $P_j(x)$ | Class-conditional distribution of $X\|Y = j$ |
| $P_j^{(k)}(x)$ | Local class-conditional distribution of $X\|Y = j$ on client $k$ |
| *4. FedReLa Core Parameters* | |
| $\rho_{\ell \to j}^{(k)}(x)$ | Re-labeling probability from local majority class $\ell$ to local minority class $j$ on client $k$ |
| $\rho_{j \to \ell}^{(k)}(x)$ | Re-labeling probability from class $j$ to $\ell$ on client $k$ (set to 0) |
| $Q^{(k)}$ | Posterior probability matrix of $D^{(k)}$ ($\|D^{(k)}\| \times C$) |
| $z_i^{(k)}$ | Class-wise z-score vector of sample $x_i^{(k)}$ on client $k$ |
| $\mu_i^{(k)}$ | Class-wise mean of posterior probabilities (for z-score) |
| $\sigma_i^{(k)}$ | Class-wise std of posterior probabilities (for z-score) |
| $t_{\text{re}}^{(k)}$ | Client-specific re-labeling threshold (tunable via $\tau$) |
| $\varpi_j^{(k)}$ | Class-wise reweighting vector (from local class priors $\pi^{(k)}$) |
| $n_Y^{(k)}$ | Class-wise sample count vector of $D^{(k)}$ |
| $\tau$ | Hyperparameter controlling re-labeling strength (top-$\tau$% z-scores) |
| *5. Imbalance & Heterogeneity* | |
| $IR(D)$ | Global imbalance ratio ($\max_j \pi_j / \min_j \pi_j$) |
| $IF$ | Imbalance factor (for long-tailed datasets) |
| $\alpha$ | Heterogeneity control parameter (Latent Dirichlet Sampling) |
| $K$ | Number of clients in the federation |
| $w_k$ | Aggregation weight of client $k$ (FedAvg: $w_k = \|D^{(k)}\|/\|D\|$) |
| *6. Decision Boundaries* | |
| $S_{j,\ell}$ | Optimal Bayesian decision boundary between classes $j$ and $\ell$ |
| $\tilde{S}^{(k)}$ | Decision boundary of client $k$ on re-labeled local dataset $\tilde{D}^{(k)}$ |

Table 3: Notation Table: Key Symbols and Definitions

**Example 1.** *We use an extreme example to illustrate that the mismatches between global and local imbalance ratios can amplify the bias in the aggregated decision boundary. Consider a binary classification problem with two classes, $j$ and $\ell$, and two clients, $k_1$ and $k_2$. Assume that the global class priors satisfy $\pi_\ell \gg \pi_j$, where $\pi_j = m_j/n$, $\pi_\ell = m_\ell/n$ and $n = |\mathcal{D}^{(k_1)}| + |\mathcal{D}^{(k_2)}|$. Here, $m_j$ and $m_\ell$, satisfying $m_j + m_\ell = n$, are the number of data points in class $j$ and $\ell$, respectively. Suppose the local dataset $\mathcal{D}^{(k_1)}$ contains $(m_j - 1)$ samples from the global minority class $j$ and one sample from the global majority class $\ell$, while the local dataset $\mathcal{D}^{(k_2)}$ contains $(m_\ell - 1)$ samples from class $\ell$ and one sample from class $j$. The local decision boundaries on $\mathcal{D}^{(k_1)}$ and $\mathcal{D}^{(k_2)}$ are:*

$$\left\{ x \in \mathcal{X} : \frac{P_j(x)}{P_\ell(x)} = \frac{1}{m_j - 1} \right\} \quad and \quad \left\{ x \in \mathcal{X} : \frac{P_j(x)}{P_\ell(x)} = m_\ell - 1 \right\}.$$

*In FL, consider the global aggregated model $\eta_j^{[w]}(x) = w_{k_1} \eta_j^{(k_1)}(x) + w_{k_2} \eta_j^{(k_2)}(x)$.*

*Suppose the aggregation weights are chosen as $w_{k_1} \propto |\mathcal{D}^{(k_1)}|$ and $w_{k_2} \propto |\mathcal{D}^{(k_2)}|$, which is widely used in imbalanced classification in the literature. Since $|\mathcal{D}^{(k_1)}| = m_j$ and $|\mathcal{D}^{(k_2)}| = m_\ell$, it follows that $w_{k_1} = \pi_j$ and $w_{k_2} = \pi_\ell$, implying $w_{k_1} \ll w_{k_2}$. In addition, the local imbalance ratios are $\mathrm{IR}(\mathcal{D}^{(k_1)}) = 1/(m_j - 1)$ and $\mathrm{IR}(\mathcal{D}^{(k_2)}) = m_\ell - 1$, so that $\mathrm{IR}(\mathcal{D}^{(k_1)}) \ll \mathrm{IR}(\mathcal{D}^{(k_2)})$. Thus, the decision boundary of the global aggregated model $\eta_j^{[w]}(x)$ is $S_{j,\ell}^{[w]} = \{x \in \mathcal{X} : P_j(x)/P_\ell(x) = \pi_\ell^{[w]}/\pi_j^{[w]}\}$ with $\pi_\ell^{[w]} = \pi_j(m_j - 1)/m_j + \pi_\ell/m_\ell$ and $\pi_j^{[w]} = \pi_j/m_j + \pi_\ell(m_\ell - 1)/m_\ell$. As a result, during model aggregation in each communication round, the global imbalance is exacerbated due to the dominant contribution from client $k_2$, amplified by both its large aggregation weight $w_{k_2}$ and local imbalance ratio $\mathrm{IR}(\mathcal{D}^{(k_2)})$.*

*Even under the uniform averaging with $w_{k_1} = w_{k_2} = 1/2$, the decision boundary of the global aggregated model $\eta_j^{[w]}(x)$ is $S_{j,\ell}^{[w]} = \{x \in \mathcal{X} : P_j(x)/P_\ell(x) = \pi_\ell^{[w]}/\pi_j^{[w]}\}$ with $\pi_\ell^{[w]} = (m_j - 1)/(2m_j) + 1/(2m_\ell)$ and $\pi_j^{[w]} = 1/(2m_j) + (m_\ell - 1)/(2m_\ell)$. The decision boundary is still biased due to the global imbalance.*

**Supplementary Explanation on Aggregated Model Representation**   To analyze how re-labeling influences the global decision boundary, we adopt the global model defined via posterior aggregation (i.e., $\sum_{k=1}^{K} w_k f(x, \theta^{(k)})$, where $f(x, \theta^{(k)})$ denotes the local posterior of client $k$ with parameter $\theta^{(k)}$) instead of parameter aggregation (i.e., $f\left(x, \sum_{k=1}^{K} w_k \theta^{(k)}\right)$). This choice is motivated by two key considerations: (1) the posterior-aggregated form renders changes in the decision boundary more explicit and easier to quantify, which aligns with our focus on analyzing re-labeling's effect; (2) it is consistent with statistical model averaging ideas, providing a flexible framework for heterogeneous FL scenarios.

No specific constraints are imposed on the aggregation weights $w_k$, and our only assumption is that the aggregated global model can be expressed as a weighted average of local posteriors, which we explicitly formalize. Furthermore, the two aggregation paradigms (parameter-aggregated and posterior-aggregated) are approximately equivalent under mild regularity conditions, as justified by first-order Taylor expansion:

Assume all local parameters $\theta^{(k)}$ are sufficiently close to a common reference value $\theta_0$ (a reasonable condition in late-stage FL training when models converge). Expanding both models around $\theta_0$:

1. For the parameter-aggregated global model:

$$f\left(x, \sum_{k=1}^{K} w_k \theta^{(k)}\right) \approx f(x, \theta_0) + \left.\frac{\partial f(x, \theta)}{\partial \theta}\right|_{\theta=\theta_0} \sum_{k=1}^{K} w_k(\theta^{(k)} - \theta_0)$$

$$= f(x, \theta_0) + \sum_{k=1}^{K} w_k \left.\frac{\partial f(x, \theta)}{\partial \theta}\right|_{\theta=\theta_0} (\theta^{(k)} - \theta_0)$$

2. For the posterior-aggregated global model (noting $\sum_{k=1}^{K} w_k = 1$):

$$\sum_{k=1}^{K} w_k f(x, \theta^{(k)}) \approx f(x, \theta_0) + \left.\frac{\partial f(x, \theta)}{\partial \theta}\right|_{\theta=\theta_0} \left(\sum_{k=1}^{K} w_k \theta^{(k)} - \theta_0\right)$$

$$= f(x, \theta_0) + \sum_{k=1}^{K} w_k \left.\frac{\partial f(x, \theta)}{\partial \theta}\right|_{\theta=\theta_0} (\theta^{(k)} - \theta_0)$$

The two expansions are identical, confirming that the parameter-aggregated and posterior-aggregated global models are **first-order equivalent** when local parameters are sufficiently close. This justifies our use of the posterior-aggregated form for analyzing decision boundary changes, as it does not introduce substantive deviations from standard parameter-aggregated FL while offering greater analytical tractability.

### A.1 PROOF OF LEMMA 2

*Proof.* As we are considering the binary classification setting, the optimal Bayesian decision boundary based on $\widetilde{\mathcal{D}}^{(k)}$ is

$$\widetilde{S}^{(k)} = \left\{ x^* \in \mathcal{X} : \widetilde{\eta}_j^{(k)}(x)(x^*) = \widetilde{\eta}_\ell^{(k)}(x^*) \right\},$$

where

$$\widetilde{\eta}_\ell^{(k)}(x) = \eta_\ell^{(k)}(x)[1 - \rho_{\ell \to j}^{(k)}(x)] + \eta_j^{(k)}(x)\rho_{j \to \ell}^{(k)}(x).$$

Given the formulation of $\widetilde{\eta}_j^{(k)}(x)$, we need

$$\eta_j^{(k)}(x^*)[1 - \rho_{j \to \ell}^{(k)}(x^*)] + \eta_\ell^{(k)}(x^*)\rho_{\ell \to j}^{(k)}(x^*) = \eta_\ell^{(k)}(x^*)[1 - \rho_{\ell \to j}^{(k)}(x^*)] + \eta_j^{(k)}(x^*)\rho_{j \to \ell}^{(k)}(x^*),$$

which is equivalent to

$$\eta_j^{(k)}(x^*)[1 - 2\rho_{j \to \ell}^{(k)}(x^*)] = \eta_\ell^{(k)}(x^*)[1 - 2\rho_{\ell \to j}^{(k)}(x^*)].$$

Regarding the fact that $\eta_j^{(k)}(x) = \pi_j^{(k)} P_j(x)/\sum_{j_0 \in \mathcal{Y}} \pi_{j_0}^{(k)} P_{j_0}(x)$, the above equation can be simplified to

$$\pi_j^{(k)} P_j(x^*)[1 - 2\rho_{j \to \ell}^{(k)}(x^*)] = \pi_\ell^{(k)} P_\ell(x^*)[1 - 2\rho_{\ell \to j}^{(k)}(x^*)],$$

and the final result follows immediately. $\qquad\square$

### A.2 PROOF OF LEMMA 3

*Proof.* The global aggregated model satisfies

$$\widetilde{\eta}_j^{[w]}(x) = \sum_{k=1}^{K} w_k \widetilde{\eta}_j^{(k)}(x) = \sum_{k=1}^{K} w_k \left\{ \eta_j^{(k)}(x) + \eta_\ell^{(k)}(x)\rho_{\ell \to j}^{(k)}(x) \right\}$$

and

$$\widetilde{\eta}_\ell^{[w]}(x) = \sum_{k=1}^{K} w_k \widetilde{\eta}_\ell^{(k)}(x) = \sum_{k=1}^{K} w_k \eta_\ell^{(k)}(x)[1 - \rho_{\ell \to j}^{(k)}(x)].$$

Then, for $x^*$ on the Bayesian decision boundary, it requires that

$$\widetilde{\eta}_j^{[w]}(x^*) = \sum_{k=1}^{K} w_k \left\{ \eta_j^{(k)}(x^*) + \eta_\ell^{(k)}(x^*)\rho_{\ell \to j}^{(k)}(x^*) \right\}$$

$$= \sum_{k=1}^{K} w_k \eta_\ell^{(k)}(x^*)[1 - \rho_{\ell \to j}^{(k)}(x^*)] = \widetilde{\eta}_\ell^{[w]}(x^*),$$

which can be simplified to

$$\sum_{k=1}^{K} w_k \eta_j^{(k)}(x^*) = \sum_{k=1}^{K} w_k \eta_\ell^{(k)}(x^*)[1 - 2\rho_{\ell \to j}^{(k)}(x^*)].$$

Applying $\eta_j^{(k)}(x) = \pi_j^{(k)} P_j(x)/\sum_{j_0 \in \mathcal{Y}} \pi_{j_0}^{(k)} P_{j_0}(x)$ again, we get the desired result. $\qquad\square$

# B ADDITIONAL EXPERIMENT DETAILS

The code is available at: `https://github.com/anonymous2025988/FedReLa.git`.

**Training details.** To ensure fair comparison, all global models are trained until full convergence with communication rounds adapted per method. Specifically, baseline methods require 500 rounds for convergence, while CLIMB, which introduces class-wise loss reweighting parameters, demands extended training: 2000 rounds on Fashion-MNIST and CIFAR-10, and 1000 rounds on CIFAR-100. As we do not intend to compare these algorithm-level methods, we use the SGD optimizer with the same weight decay and momentum as they reported in their original implementations: weight decay of $0.00001$ and momentum $0.9$ for all methods except long-tailed-oriented methods FedETF and FedLOGE, which follow their original implementations with zero weight decay and momentum $0.5$. All experiments were conducted with three distinct random seeds, and their average results are reported in the tables.

**Evaluation metrics.** A balanced test dataset is used to evaluate the overall accuracy performance of the global model. Additionally, the average test accuracy for both minority and majority classes is reported for the step-wise imbalanced setting. For long-tailed datasets, we report the accuracy over head, medium, and tail classes as Many-, Medium-, and Few-shot, respectively.

Adhering to the long-tailed federated learning protocol established in (Xiao et al., 2024), we categorize classes into three disjoint subsets based on sample size distribution: head (majority), medium, and tail (minority) class groups, constituting 75%, 20%, and 5% of total samples, respectively. For long-tailed CIFAR-10, we define classes $\{0, 1, 2\}$ as head classes, $\{3, 4, 5\}$ as medium classes, and $\{6, 7, 8, 9\}$ as tail classes. On long-tailed CIFAR-100, this partitioning extends with classes 0-47 forming the head partition, 48-83 as medium, and 84-99 as tail. To evaluate model performance through stratified accuracy metrics, we report Head/Medium/Tail-shot accuracies corresponding to these partitions in Table 2.

## B.1 COMPUTATIONAL COST

All experiments were conducted on a Spartan cluster on a single node equipped with one NVIDIA H100 GPUs (80GB memory) 10GB RAM with 12 CPU cores.

Before approximating the computational cost of FedReLa, we would like to clarify the fundamental difference between **extra Local training** and **extra local computation**:

1. Local training overhead involves **gradient updates for new parameters or module**. For example, methods that introduce new optimizable parameters (e.g., CLIMB, FedLOGE, etc.) require extra per-round local training overhead to update the gradients of these parameters.

2. ONE-TIME Model Inference: FedReLa only performs **one-time model inference during a single round** to obtain posterior probabilities, without updating the model or gradients. Therefore, we describe FedReLa as operating "without extra local training."

**Approximate one-time computation cost of FedReLa.** The strength of FedReLa as a data-level method lies in its requirement for only a single computational step during a single round to refine the imbalanced data distribution, thereby achieving long-lasting improvements in model performance. The core operation of FedReLa is model inference (forward pass) to obtain the local posterior probability matrix $Q^{(k)}$. We approximately consider

$$\text{FLOP}_{\text{train}} = \text{FLOP}_{\text{forward}} + \text{FLOP}_{\text{backpropagation}},$$
$$\text{FLOP}_{\text{backpropagation}} \approx 2 \times \text{FLOP}_{\text{forward}}.$$

Thus, the FLOPs required for $Q^{(k)} = f(\theta_{T_{\text{relabel}}}^{\text{global}}; X^{(k)})$ can be approximatly quantified with:

$$\text{FLOP}_{Q^{(k)}} = \text{FLOP}_{\text{forward}} \approx \frac{1}{3}\text{FLOP}_{\text{train}}.$$

The total computation cost of FedReLa is approximately 1/3 of the computation cost of a single training round. This cost is One-Time only during the single round of $T_{\text{relabel}}$.

**Runtime comparison.** Unlike methods requiring from-scratch training, FedReLa enhances classifier performance solely through one-shot re-labeling during the fine-tuning phase. Consequently, its computational overhead is primarily determined by the base federated learning algorithm it augments. For instance, each communication round of FedLOGE requires an average of 72.36 seconds on CIFAR100. When FedReLa enhances FedLOGE with a one-shot computation for label re-allocator, the average communication round time increased to 73.06 seconds, which is negligible.

B.2 ADDITIONAL EXPERIMENT RESULTS

**Additional experiment on step-wise setting with resent SOTAs.** Although recent methods, such as FedETF (Li et al., 2023) and FedLOGE (Xiao et al., 2024), are long-tail-oriented approaches, we conducted additional experiments on CIFAR-10 with step-wise imbalance. The results in Table 4 demonstrate that FedReLa still achieves SOTA performance on step-wise imbalance. FedReLa brings significant improvements, especially under higher imbalance ratios and more heterogeneous data.

| | | $\alpha = 0.3$ | | $\alpha = 0.1$ | |
|---|---|---|---|---|---|
| IR | Method | Minority/Majority | Overall | Minority/Majority | Overall |
| 10 | FedETF | 74.21/93.79 | 84.01 | 44.14/96.46 | 70.30 |
| | **+FedReLa** | 82.11/91.72 | 86.92 | **68.28/84.52** | **76.43** |
| | FedLOGE | 80.49/92.12 | 86.32 | 61.57/86.05 | 73.81 |
| | **+FedReLa** | **85.81/89.73** | **87.77** | 68.7/81.64 | 75.17 |
| 20 | FedETF | 69.31/87.73 | 78.52 | 43.65/72.55 | 58.10 |
| | **+FedReLa** | **82.75/83.74** | **83.11** | **67.85/75.75** | **71.82** |
| | FedLOGE | 79.30/85.9 | 82.60 | 45.61/63.79 | 54.70 |
| | +FedReLa | 79.60/84.21 | 81.90 | 59.18/70.73 | 64.95 |

Table 4: Performance on Step-wise-imbalanced CIFAR10

**Additional experiment on higher proportion of minority classes.** In addition to 10% and 30% minority classes for step-wise-imbalanced datasets, we further extend the proportion to 50% to examine the consistency of enhancement from FedReLa on extreme conditions. As demonstrated in Table 5, FedReLa delivers significant performance gains even in the extreme case where minority classes constitute 50% of the data. Without the FedReLa boost, baseline methods exhibit pronounced accuracy degradation as the proportion of the minority class increases. Our proposed label re-allocator effectively mitigates this performance deterioration while simultaneously enhancing overall accuracy. These results strongly validate FedReLa's capability to provide robust performance enhancements for federated learning methods that face substantial minority class presence.

| | IR=10 with 50% Minority Classes | | | | | |
|---|---|---|---|---|---|---|
| | Fashion-MNIST | | CIFAR-10 | | CIFAR-100 | |
| Method | Minority | Overall | Minority | Overall | Minority | Overall |
| FedAvg | 63.44(77.10)**+13.66** | 78.29(82.90)**+4.61** | 16.94(44.42)**+27.48** | 48.67(56.72)**+8.05** | 12.17(25.27)**+13.10** | 42.48(46.76)**+4.28** |
| FedProx | 63.78(77.24)**+13.46** | 78.62(82.62)**+4.00** | 16.32(43.54)**+27.22** | 48.20(56.52)**+8.32** | 12.83(23.13)**+10.30** | 41.97(46.28)**+4.31** |
| FedNova | 77.60(80.50)**+2.90** | 81.89(83.30)**+1.41** | 23.47(39.20)**+15.73** | 52.70(55.65)**+2.95** | 13.53(23.23)**+9.70** | 45.70(47.57)**+1.87** |
| MOON | 64.68(76.60)**+11.92** | 79.22(82.74)**+3.52** | 11.20(38.54)**+27.34** | 46.35(54.69)**+8.34** | 11.17(24.23)**+13.06** | 42.53(46.56)**+4.03** |
| CLIMB | 75.47(79.90)**+4.43** | 85.22(87.14)**+1.92** | 31.35(42.85)**+11.50** | 69.68(72.01)**+2.33** | 11.00(24.48)**+13.48** | 30.71(34.79)**+4.08** |

Table 5: Test accuracies (in %) of different methods on step-wise imbalance datasets under IR=10 with 50% minority classes at heterogeneity level $\alpha = 0.3$ in the format of `original(+FedReLa)+`**`enhancement`**
.

**Large-Scale Dataset Validation on ImageNet-LT.** We conduct supplementary experiments on ImageNet-LT with data heterogeneity level $\alpha = 0.1$, 20 clients, and 0.4 participation fraction. Another recent SOTA method, FedYoYo (Yan et al., 2025), is used to demonstrate the algorithmic agnosticism of FedReLa. Although FedYoYo requires each client to upload the estimated local distribution for aggregation on the server, making it fall outside the scope of our core comparisons (raises concerns on data privacy in federated learning), we still include this experiment to demonstrate that FedReLa can consistently enhance performance even when integrated with methods that rely

on extra privacy-sensitive information. We focus on evaluating whether FedReLa can maintain performance gains as it adapts to large-scale data distributions.

| Method | Overall Accuracy (%) | H/M/T Accuracy (%) |
|---|---|---|
| FedYoYo (Yan et al., 2025) | 38.15 | 41.19/39.42/31.08 |
| +FedReLa | **38.78** | 40.73/**40.06/33.71** |
| FedLoGe (Xiao et al., 2024) | 30.52 | 46.29/28.01/15.02 |
| +FedReLa | **31.70** | 45.43/**30.44/18.02** |

Table 6: Performance comparison on ImageNet-LT (H=Head, M=Medium, T=Tail). FedReLa enhances tail-class accuracy by 2.63% without sacrificing overall performance.

The result above confirms that FedReLa's sample-level re-labeling mechanism—calibrated via classwise z-score standardization—avoids the scalability bottlenecks of feature-space methods (e.g., SMOTE) and maintains effectiveness on large-scale datasets. The consistency of performance gains validates FedReLa's inherent scalability for real-world large-scale federated learning scenarios.

**Large number of clients on CIFAR100-LT** We extend CIFAR100-LT experiments to 50 and 100 clients, with an imbalance factor $imb\_factor = 100$ and data heterogeneity $\alpha = 0.1$. The participation fractions are set to 0.2 and 0.1, respectively. As the number of clients increases, class absences become increasingly severe. This setup aims to verify FedReLa's robustness to extreme class imbalance and its compatibility with diverse algorithmic approaches.

| Method | 50 Clients | | 100 Clients | |
|---|---|---|---|---|
| | Overall Accuracy (%) | H/M/F Accuracy (%) | Overall Accuracy (%) | H/M/F Accuracy (%) |
| FedLC (Zhang et al., 2022) | 32.81 | 56.20/32.42/9.74 | 23.72 | 46.41/20.74/4.23 |
| +FedReLa | **34.76** | 49.92/**35.24/17.21** | **25.13** | 39.82/**28.34/7.02** |
| FedYoYo (Yan et al., 2025) | 40.89 | 54.32/41.95/24.12 | 30.73 | 34.12/29.64/27.92 |
| +FedReLa | **41.31** | 53.62/**42.24/25.91** | **32.13** | 33.90/**32.72/29.22** |
| FedETF | 31.89 | 60.67/40.45/9.22 | 28.71 | 59.00/33.72/9.73 |
| +FedReLa | **33.94** | 55.12/**43.61/15.12** | **31.50** | 52.21/**38.73/16.53** |
| FedLoGe | 34.83 | 57.12/42.21/17.29 | 33.08 | 62.71/38.78/13.75 |
| +FedReLa | **35.62** | 57.01/**44.32/18.00** | **34.32** | 59.90/**42.22/16.00** |

Table 7: Performance comparison on CIFAR100-LT with 50/100 clients (H=Head, M=Medium, F=Few-shot). FedReLa consistently boosts few-shot accuracy across baselines.

Table 7 demonstrates three key conclusions: (1) FedReLa delivers consistent enhancements for all baselines; (2) Even with 100 clients (a large-scale client setup), FedReLa maintains performance gains, validating its scalability to distributed environments with numerous clients and its robustness to severe class absence; (3) The consistent improvements across diverse algorithmic paradigms further confirm FedReLa's algorithm-agnostic property as a data-level plug-in.

**Tail-Class Accuracy Gain on CIFAR10-LT Under Extreme Heterogeneity** Under $\alpha = 0.1$, 70% of tail classes are absent from clients, simulating extreme real-world heterogeneity.

| Method | Imbalance Factor (IF) | Tail-Class Accuracy Gain (%) | | |
|---|---|---|---|---|
| +FedReLa | | $\alpha = 0.1$ | $\alpha = 0.3$ | $\alpha = 10$ |
| FedETF | 50 | **+36.49** | +5.52 | +3.68 |
| FedETF | 100 | **+30.70** | +10.50 | +2.51 |
| FedLoGe | 50 | **+14.97** | +3.43 | +5.76 |
| FedLoGe | 100 | **+19.73** | +9.58 | +4.40 |

Table 8: Tail-class accuracy gain of FedReLa on CIFAR10-LT. Improvements are more significant under extreme heterogeneity ($\alpha = 0.1$).

As presented in Table 8, FedReLa's tail-class gains are most pronounced under extreme heterogeneity ($\alpha = 0.1$). In contrast, gains are smaller under mild heterogeneity ($\alpha = 10$, close to IID). This result confirms that FedReLa is designed explicitly for heterogeneous scenarios with severe class absence: its deferred re-labeling strategy and calibrated posterior estimation enable effective identification of majority-class samples similar to absent minorities, avoiding blind re-labeling and delivering robust performance gains.

### B.3 SENSITIVITY ANALYSIS

We perform the sensitivity analysis of re-labeling threshold $t_{\mathrm{re}}^{(k)}$ on long-tailed CIFAR-10 with IF = 50. On each client, the class-wise threshold $t_{\mathrm{re}}^{(k)}$ is determined by the top-$\tau$ % of z-scores. The threshold $t_{\mathrm{re}}^{(k)}$ controls the re-labeling strength (the amount of re-labeled samples) as demonstrated in Figure 2(a). This serves as a safeguard to regulate the number of samples re-labeled by FedReLa. For instance, using the top 1% z-score as the re-labeling threshold limits the number of re-labeled samples to be less than 1% of local data. Figure 2(a) shows that the amount of re-labeled samples scales linearly with top-$\tau$ percentiles.

In Figure 2(b), when $\tau \leq 5$, tail-class performance gains outweigh head-class losses. When $\tau > 5$, medium-class accuracy steadily improves and head-class accuracy continues to decline slowly, while tail-class accuracy remains relatively stable. The effect on performance of turning $t_{\mathrm{re}}^{(k)}$ up reveals that: (1) Initially, re-labeled head-class samples with a small $\tau$ mostly invade tail-class feature space. (2) After re-labeling these critical samples, further label re-allocating relieves the head-class invasion of the medium-class feature space. (3) FedReLa prioritizes re-labeling samples that most severely invade tail-class regions. We observe similar results on the CIFAR100-LT (Table 9 in the appendix).

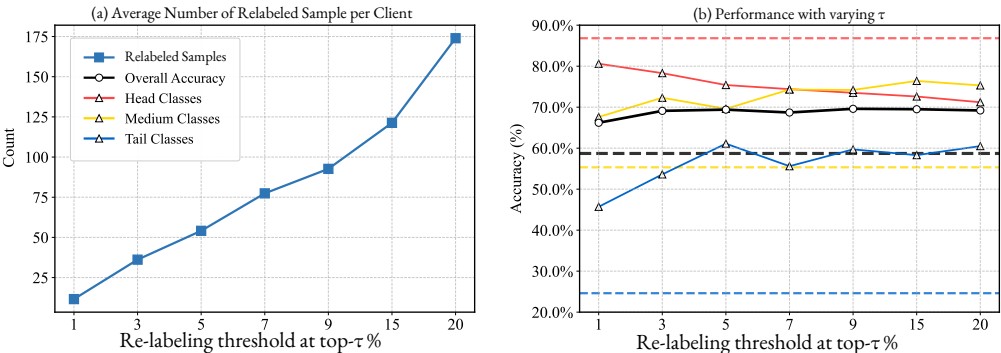

Figure 2: Sensitive analysis respect to $\tau$, which controls the re-labeling strength.

The threshold-tuning capability allows FedReLa to deliver customized class-wise enhancement, prioritizing tail-class gains ($\tau = 5$) while preserving overall performance. This strategic trade-off (suppressing overprivileged head classes to boost tails) is a unique advantage over static algorithm-level approaches (Li et al., 2023; Xiao et al., 2024), as evidenced by the accuracy curves surpassing the baseline (dashed lines) in critical regions. In practice, we can tune the trade-off through $\tau$ depending on how much importance we place on minority-class performance.

Recall the conclusion from observations on CIFAR-10-LT: (1) Initially, re-labeled head-class samples with a small $k$ mostly invade tail-class feature space. (2) After re-labeling these critical samples, further label re-allocating relieves the head-class invasion of the medium-class feature space. (3) FedReLa prioritizes re-labeling samples that most severely invade tail-class regions. We observe similar results on the CIFAR100-LT dataset, which are presented in Table 9. We anticipate that the optimal parameters will exhibit slight differences across datasets with varying posterior probability distributions and degrees of class overlap. When $\tau = 3$, FedReLa achieves maximum performance gain on CIFAR-100-LT, where the degree of class overlap is more severe. Although the parameter range 1-20% consistently provides performance gain on both CIFAR-10 and CIFAR-100, with the principle of minimizing data-editing, **we recommend using slightly conservative relabeling strength (3%-5%).**

Table 9: Sensitivity analysis on CIFAR-100-LT

| top-$\tau$ % | Original | 1 | 3 | 5 | 7 | 9 | 15 | 20 |
|---|---|---|---|---|---|---|---|---|
| Overall | 44.1 | 44.6 | **45.1** | 44.7 | 44.6 | 44.9 | 44.5 | 44.6 |
| Many-shot | 56.4 | 57.2 | 58.5 | 56.6 | 57.1 | 56.8 | 56.2 | 56.0 |
| Medium-shot | 49.2 | 49.5 | 48.8 | 50.0 | 49.8 | 49.4 | 50.7 | 51.1 |
| Few-shot | 26.6 | 27.4 | 28.4 | 28.0 | 27.5 | 28.9 | 27.6 | 26.8 |
| Relabeled | 0 | 51 | 157 | 194 | 216 | 276 | 317 | 364 |

The threshold-tuning capability enables FedReLa to deliver customized class-wise enhancements, prioritizing tail-class gains while preserving overall performance. This strategic trade-off (suppressing overprivileged head classes to boost tails) is a unique advantage over static algorithm-level approaches (Li et al., 2023; Xiao et al., 2024), as evidenced by the accuracy curves surpassing the baseline (dashed lines) in critical regions. Again, in practice, we can tune the trade-off through $t_{\text{re}}^{(k)}$ by $\tau$ depending on how much importance we place on minority-class performance.

## C  ABLATION STUDY

**Ablation study on the importance of Z-score standardization**  Z-score standardization is critical for enabling FedReLa to utilize the underestimated posterior probabilities output by biased models. To validate its necessity, we conducted ablation experiments on CIFAR-10-LT ($\alpha = 0.1$, IF = 50) without standardization, and directly using posterior probabilities as flip probabilities.

| Method | Overall | Many-shot | Medium-shot | Few-shot |
|---|---|---|---|---|
| FedLOGE | 57.5 | 83.0 | 61.1 | 19.8 |
| +FedReLa | **70.0** | 76.0 | **72.7** | **59.4** |
| +FedReLa w/o Z-score | 59.7 | 82.1 | 72.3 | 17.4 |

Table 10: Performance of FedReLa with/without Z-score Standardization

Without z-score standardization, the Few-shot performance fails to show improvement. This is attributed to the fact that the posterior probabilities are underestimated by the biased global model for tail classes and are typically extremely small. Directly utilizing them as flipping probabilities hinders the effective conversion of these samples into global minority classes. Meanwhile, the Medium-shot performance exhibits improvement as these classes possess more samples than tail classes, resulting in the model underestimating their posterior probabilities to a lesser extent. Thus, head-class samples with similar features are preferentially flipped to the medium class, rather than to the tail classes with tiny posterior probabilities.

Applying z-score standardization to the underestimated posterior probabilities enables a balanced label re-allocating behavior, which achieves a better balanced trade-off among the performance of Head, Medium, and Tail classes. Ultimately, this contributes to the superior Overall accuracy.

**Ablation study on data-heterogeneity.**  To evaluate FedReLa's performance under higher imbalance ratios across varying degrees of data heterogeneity, we increased the imbalance ratio (IR) to 20 and the number of clients to $K = 100$ on the Fashion-MNIST dataset with 3 minority classes.

Results in Table 11 depict consistent performance improvements by FedReLa across different levels of data heterogeneity on the Fashion-MNIST dataset for each algorithm-level method. This highlights the robustness of FedReLa in mitigating the impact of data heterogeneity through enhancements to both data and classifiers.

Improved percentage shows that the improvement achieved by FedReLa increases with higher data heterogeneity, indicating that FedReLa-boosted models exhibit significantly improved robustness to heterogeneous data distributions compared to baseline methods.

As $\alpha$ decreases (i.e., heterogeneity increases), FedReLa demonstrates progressively greater improvements in both minority-class accuracy (+8.13% to +35.40%) and overall accuracy (+1.83%

| | Original (+FedReLa) Performance | | Improved Percentage (%) | |
|---|---|---|---|---|
| | Minority Accuracy | Overall Accuracy | Minority Accuracy | Overall Accuracy |
| | $\alpha = 10$ | | | |
| FedAvg | 51.70 (81.83) | 79.57 (84.56) | +30.13 | +4.99 |
| FedProx | 51.57 (81.83) | 79.35 (84.40) | +30.26 | +5.05 |
| FedNova | 52.00 (82.27) | 79.35 (84.61) | +30.27 | +5.26 |
| MOON | 45.77 (81.17) | 78.35 (85.60) | +35.40 | +7.25 |
| CLIMB | 55.80 (69.57) | 82.39 (86.14) | +13.77 | +3.75 |
| | $\alpha = 5$ | | | |
| FedAvg | 45.27 (78.53) | 77.80 (84.42) | +33.26 | +6.62 |
| FedProx | 45.57 (78.03) | 77.90 (84.27) | +32.46 | +6.37 |
| FedNova | 46.10 (78.60) | 78.18 (84.30) | +32.50 | +6.12 |
| MOON | 45.57 (79.73) | 78.21 (85.51) | +34.16 | +7.30 |
| CLIMB | 56.10 (69.53) | 82.67 (86.35) | +13.43 | +3.68 |
| | $\alpha = 1$ | | | |
| FedAvg | 50.67 (75.57) | 79.06 (84.12) | +24.90 | +5.06 |
| FedProx | 50.10 (74.60) | 78.95 (84.25) | +24.50 | +5.30 |
| FedNova | 50.10 (75.93) | 78.91 (84.47) | +25.83 | +5.56 |
| MOON | 44.87 (74.97) | 77.94 (84.95) | +30.10 | +7.01 |
| CLIMB | 61.17(69.3) | 83.78(85.93) | +8.13 | +2.15 |
| | $\alpha = 0.3$ | | | |
| FedAvg | 50.50 (75.70) | 78.46 (83.84) | +25.20 | +5.38 |
| FedProx | 50.00 (74.90) | 78.48 (83.63) | +24.90 | +5.15 |
| FedNova | 55.03 (77.90) | 76.67 (84.46) | +22.87 | +7.79 |
| MOON | 44.43(74.77) | 77.12(84.47) | +30.34 | +7.35 |
| CLIMB | 53.43 (64.70) | 81.42 (84.56) | +11.27 | +3.14 |
| | $\alpha = 0.1$ | | | |
| FedAvg | 33.83 (67.60) | 68.43 (79.25) | +33.77 | +10.82 |
| FedProx | 34.40 (68.27) | 69.37 (79.35) | +33.87 | +9.98 |
| FedNova | 70.43 (82.83) | 74.25 (82.10) | +12.40 | +7.85 |
| MOON | 22.59 (45.82) | 67.44 (77.88) | +23.23 | +10.44 |
| CLIMB | 56.97 (65.20) | 81.78 (83.61) | +8.23 | +1.83 |

Table 11: Ablation study on $\alpha$. The overall accuracy and average accuracy of minority classes (in %) on step-wise Fashion-MNIST with 3 minority classes (30%) for IR = 20 with 100 clients. The results in brackets show the FedReLa enhanced performance.

to +10.82%), with the most significant gains observed under extreme non-IID scenarios ($\alpha = 0.1$). While baseline methods exhibit varied sensitivity to heterogeneity, CLIMB shows inherent robustness but limited enhancement headroom, and MOON suffers significant performance drops at $\alpha = 0.1$. Yet FedNLR consistently mitigates these limitations through adaptive calibration, offering consistent enhancement. Notably, FedReLa reduces minority-class accuracy disparities by 23-37% across $\alpha \leq 1$ while maintaining global model stability, particularly excelling in balancing the accuracy tradeoff between dominant and rare classes. These results position FedReLa as a versatile solution for real-world federated learning deployments, offering three key advantages: 1) enhanced robustness to severe data heterogeneity without requiring client-specific tuning, 2) compatibility with existing aggregation frameworks, and 3) simultaneous optimization of both class-balanced and global model performance in non-IID environments.

