# OpenReview forum: "FedReLa: Imbalanced Federated Learning via Re-Labeling"
_ICLR.cc/2026/Conference — Submitted to ICLR 2026_

### Official Review · Reviewer_A4Ud · 2025-10-30

**Soundness:** 2
**Presentation:** 3
**Contribution:** 3
**Rating:** 6
**Confidence:** 4

**Summary:**

This paper targets a realistic FL setting where global class imbalance coexists with cross-client heterogeneity. It proposes a data-level “one-shot label reallocation” approach (FedReLa). Without relying on global priors or extra communication, the method uses the global model’s local posteriors, within-class z-score normalization, and a tanh thresholding step to identify “suspicious majority-class samples” near minority-class regions and probabilistically relabels them as minority. The goal is to collectively “push back” biased decision boundaries at both local and aggregated levels to mitigate imbalance. The paper claims plug-and-play integration, very low overhead, and composability with algorithm-level methods.

**Strengths:**

- The paper focuses on a more realistic and challenging combination in FL: global imbalance + heterogeneity + local-global mismatch. This is a relevant and important problem.
- The solution is data-level, requires no global priors or auxiliary data, and can be easily integrated into existing FL pipelines—practically attractive.
- The one-shot local forward pass is low-cost, introduces no extra trainable parameters or communication, and is thus deployment-friendly.

**Weaknesses:**

- The paper models the aggregated global posterior as a weighted sum of local posteriors, whereas FedAvg averages parameters, and neural-network posteriors are not linearly additive. Deriving the aggregated decision boundary via “posterior averaging” is not rigorous. The authors should justify or correct the assumption that the global posterior equals a weighted sum of local posteriors. If this assumption does not hold, do Lemma 3 and the aggregated-boundary analysis still stand? Please provide empirical evidence quantifying the discrepancy between predictions from parameter-averaged models and those from weighted averaging of local predictions.
-  The theoretical analysis appears restricted to binary classification. Can it be extended to the multi-class case? The theory adopts a Bayesian decision-boundary perspective: strategic label reallocation is equivalent to adjusting effective prior ratios, which pulls back biased boundaries at local/global levels and improves minority/tail recognition—purportedly without global priors or extra communication. Related ideas appear in [1][2]; what are the precise differences and advantages of this work compared to [1][2]? Please add a focused discussion. Also, FedETF seems to pursue a similar effect; why is it meaningful to use FedReLa jointly with FedETF rather than redundantly?
-  Appendix A’s Example 1 is vague. Please clarify what, exactly, the example is intended to illustrate. Also, in FL, one may use uniform averaging rather than weighted averaging—how would that change the conclusions?
-  Despite the theoretical discussion, the proposed method seems essentially heuristic. Please clarify the precise relationship between the method and the theory—what parts of the method are directly justified by the theory, and what parts are heuristic design choices?
- Why use independent Bernoulli sampling rather than a single Categorical draw? If multiple ones occur and you then take argmax, does this introduce bias?
- Why prefer label reallocation over approaches such as SMOTE-like interpolation to synthesize new samples? What advantages does FedReLa offer relative to such feature-space/data-space augmentation methods under FL constraints?
- The paper does not appear to provide code. Could the authors release reproducible code?

[1] Federated Learning with Label Distribution Skew via Logits Calibration

[2] Aligning model outputs for class imbalanced non iid federated learning

**Questions:**

Refer to Weaknesses.

---

> ### Author Response · Authors · 2025-11-17
> **Addressing Concerns of Reviewer A4Ud (Part 1)**
>
> ### **Addressing W1: Aggregated posterior assumption**
>
> Thank you for your insightful comment. You are correct that the global model obtained by aggregating parameters differs from that formed by aggregating posterior distributions. In our work, we focus on examining how relabeling influences the decision boundary. To investigate this influence, **adopting the global model defined via posterior aggregation is more convenient, as such a representation renders changes in the decision boundary more explicit and easier to analyze**. This choice also aligns with ideas from statistical model averaging.
>
> We would further clarify that we do not impose specific constraints on aggregation weights. Our only assumption is that the aggregated model can be expressed in this posterior-aggregated form. *We agree with your observation and will explicitly formalize this assumption in the revision.*
>
> On the other hand, the two types of global models (parameter-aggregated and posterior-aggregated) can be closely related under some conditions. Suppose the local posterior takes the form $f(x, \theta^{(k)})$, where $f$ denotes a specific model structure, and $\theta^{(k)}$ represents the local parameter of client $k$. The parameter-aggregated global model can be written as $f\left(x, \sum_{k=1}^{K} w_k \theta^{(k)}\right)$, while the posterior-aggregated global model is expressed as $\sum_{k=1}^{K} w_k f(x, \theta^{(k)})$.
>
> Assume all local parameters $\theta^{(k)}$ are close to a common value $\theta_0$. A first-order Taylor expansion around $\theta_0$ yields:
>
> $$
> \\begin{aligned}
> f\\left(x, \\sum_{k=1}^{K} w_k \\theta^{(k)}\\right) &\\approx f(x,\\theta_0) + \\frac{\\partial f(x,\\theta)}{\\partial\\theta}\\vert_{\\theta=\\theta_0} \\sum_{k=1}^{K} w_k (\\theta^{(k)} - \\theta_0) \\\\
> &= f(x,\\theta_0) + \\sum_{k=1}^{K} w_k \\frac{\\partial f(x,\\theta)}{\\partial\\theta}\\vert_{\\theta=\\theta_0} (\\theta^{(k)} - \\theta_0)
> \\end{aligned}
> $$
>
> Similarly, for the posterior-aggregated model (given that $\sum_{k=1}^{K} w_k = 1$):
>
> $$
> \\begin{aligned}
> \\sum_{k=1}^{K} w_k f(x, \\theta^{(k)}) &\\approx f(x,\\theta_0) + \\frac{\\partial f(x,\\theta)}{\\partial\\theta}\\vert_{\\theta=\\theta_0} \\left(\\sum_{k=1}^{K} w_k \\theta^{(k)} - \\theta_0\\right) \\\\
> &= f(x,\\theta_0) + \\sum_{k=1}^{K} w_k \\frac{\\partial f(x,\\theta)}{\\partial\\theta}\\vert_{\\theta=\\theta_0} (\\theta^{(k)} - \\theta_0)
> \\end{aligned}
> $$
>
> Thus, under mild regularity conditions and when the local parameters $\theta^{(k)}$ are sufficiently close, the **two global models are approximately equivalent to first order**. This provides further justification for representing the aggregated model as a weighted average of posterior distributions.
>
> ### **Addressing W2a: Extension to multiclass classification**
> 1. We adopt binary classification mainly because FedReLa relabels samples from its original class to one minority class with the highest similarity. Focusing on the decision boundary between two classes clearly demonstrates FedReLa’s core principle and avoids unnecessary complexity from multiclass interactions that might obscure the key mechanism.
> 2. Such class-pairwise ($\ell \to j$) re-labeling between each majority-minority pair can be naturally extended to multiclass scenarios.
> 3. Experiments on multiple multiclass datasets (e.g., minority-class accuracy improves by 15.04% on CIFAR-100 (100 classes, IR=20)) confirm this extension. As future work, we plan to generalize the theoretical extension of FedReLa to multiclass tasks.
>
> ### **Addressing W2b: Advantages over [1][2] and compatibility with FedETF**
>
> 1. FedReLa shares a similar theoretical starting point with most logit calibration methods like [1] and FedLoGe, but differs fundamentally in its theoretical motivation: it balances decision boundaries at the data level, whereas these counterparts rely on algorithmic-level improvements (e.g., loss or architecture engineering) to learn better classifiers within limited data distributions.
> 2. Algorithm-level methods face an upper performance bound constrained by extremely imbalanced or class-absent training data—they cannot expand the minority class itself. For example, FedLC adjusts logits using local class priors, yet when the minority class is entirely absent from the local dataset, its logits cannot be effectively adjusted.
> 3. **FedReLa aims to break this limitation** by relabeling majority-class samples to balance local data and enrich the minority class’s feature representation, thereby providing better training data for these algorithm-level methods.
> 4. Experiments comparing these methods aim to verify two key points:
>    - FedReLa consistently improves data quality,
>    - and it is complementary to algorithmic-level methods.
>
> Critically, FedReLa acts as a model/algorithm-agnostic plug-in module with no extra communication cost, seamlessly enhancing algorithm-level methods. This highlights its unique advantage in practical FL deployments.

---

> ### Author Response · Authors · 2025-11-18
> **Addressing Concerns of Reviewer A4Ud (Part 2)**
>
> ### **Addressing W3:**
> 1. Example 1 is designed to illustrate why standard FL with weighted averaging amplifies bias toward the global majority class $\ell$. Weighted averaging ($w_k \propto |D^{(k)}|$) assigns far higher weights to $\ell$-dominated clients (large sample size $m_\ell$) than $j$-dominated clients, and exacerbates the extreme $\ell$-biased local decision boundaries that dominate the global model.
>
> 2. For uniform averaging $(w_{k1}=w_{k2}=1/2$), the bias by class imbalance persists, though without being exacerbated by imbalanced aggregation weights.  For example, with the same learning rate and batch size,  local gradient updates are still imbalanced as $\ell$-dominated clients have far more training batches than minority class clients, leading to strongly $\ell$-biased gradient updates. However, equal weighting can lead to underperforming models excessively influencing the performance of well-trained local models, thereby degrading the performance of the majority class.
>
> 3. Thus, weighted aggregation based on FedAvg is widely used in imbalanced classification in the literature. Example 1 aims to use this commonly adopted aggregation method to motivate the necessity of our approach.
> **We have revised Example 1 by allowing more general choices of aggregation weights (including uniform) and incorporating the above discussions.**
>
> ### **Addressing W4:  Theory and heuristic design**
>
> Following the theoretical motivation of re-labeling’s impact on decision boundaries, the components of FedReLa are guided and justified by theoretical goals, with a heuristic design to address practical issues. Below, we elaborate on the heuristic design and theoretical justification for each component:
> 1. Extracting the posterior probability matrix $Q$: Justified by the relationship $\rho \propto \eta_{l \to j}$, where $\eta_{l \to j}$ denotes the posterior probability that a sample from the majority class $l$ is classified as the minority class $j$.
> 2. Class-wise Z-score standardization: Due to the underestimation of posterior probabilities for minority classes by biased global models, standardization is required to eliminate this bias. The choice of Z-score standardization is a heuristic design, empirically validated. The class-wise formulation is derived from the theoretical analysis of class-pairwise decision boundaries, ensuring independent standardization for each class.
> 3. Local class prior weight $\varpi$: Verified by Equation 2 in Remark 1, this component ensures the shift of decision boundaries towards the correct direction induced by asymmetric re-labeling.
> 4. tanh normalization: The adoption of tanh for converting Z-scores to probabilities serves the theoretical goal of confining the re-labeling probability $\rho$ within the range [0, 1]. It is also a heuristic design that incorporates $\tau$—a tunable safeguard parameter.
> 5. Hyperparameter $\tau$: Since the obtained Z-scores essentially represent the ranking of candidate samples’ similarity to the target class, a class-wise threshold is heuristically set to filter out samples with insufficient similarity.
>
> FedReLa is motivated by the theoretical analyses in Lemmas 1–3. Its heuristic designs are guided by theoretical objectives and supported by empirical verification.
>
> ### **Addressing W5: Bernoulli v.s Categorical draw**
>
> Thank you for this insightful question. Our understanding of a "single Categorical draw" is that it directly assigns a target class to a sample based on class-wise re-labeling probabilities $\rho$ (please correct us if this understanding is inaccurate). The reason we opt for independent Bernoulli sampling stems from our independent analysis of class-pair decision boundaries. Through independent sampling, we ensure that candidate samples for each target class are independent of other classes.
>
> We address two specific scenarios as follows:
>    - A sample cannot be a candidate for any class due to excessively low re-labeling probabilities. This indicates that the sample is highly discriminable from other classes, so we do not intend to relabel it. In contrast, a single Categorical draw forces assignment to a class, leading to over-relabeling.
>    - A sample becomes a candidate for multiple classes because the $\rho$ values for these target classes are non-zero. This means the sample lies at the margins of multiple classes. Thus, we use argmax to select the class it is most similar to (i.e., the one with the highest $\rho$) as the target class. The Categorical draw cannot guarantee that the most similar class is selected as the target, potentially leading to inaccurate labeling.
>
> FedReLa normalizes all classes to a consistent scale via class-wise Z-score standardization and class-wise thresholds. Therefore, we believe the argmax operation based on this normalization does not introduce excessive bias. This conclusion is also supported by our experimental results: both the tail and the medium classes show improvements to varying degrees.

---

> ### Author Response · Authors · 2025-11-19
> **Addressing Concerns of Reviewer A4Ud (Part 3)**
>
> ### **Addressing W6: Advantage over feature-space/data-space augmentation methods**
>
> This is a critical question, and it is precisely where FedReLa’s contribution lies. As discussed in the "Related Works" section (Lines 96–107) of the paper, methods that synthesize new samples (such as SMOTE) have two key limitations:
> 1. **Lack of seed samples**: Local class absence caused by data heterogeneity is highly prevalent. SMOTE or feature-space/data-space augmentation methods fail to synthesize new samples because they lack the necessary seed samples/features.
> 2. **Unknown global data distribution**: This prevents these methods from identifying which classes require augmentation.
>
> **As the first data-level method for federated learning, FedReLa addresses the issues posed by class absence and unknown global distribution through re-labeling**. It implements a lightweight plug-in approach, significantly improving the model’s performance on minority classes without relying on global distribution information and with zero additional communication overhead.
>
> ### **Addressing W7: Code release**
>
> Thank you for raising this point. The code now is available here: https://github.com/anonymous2025988/FedReLa.git
>
> ### **Supplementary experimental results on FedLC [1]**
> We have conducted experiments to show the joint performance of FedReLa and the logits-adjustment method FedLC [1] (algorithm-level method) on Long-tailed CIFAR-100 (imb_factor = 100, α = 0.1) with 50 and 100 clients.
>
> | CIFAR-100-LT | 50 Clients |  | 100 Clients |  |
> |:---:|:---:|:---:|:---:|:---:|
> |  | Overall | H/M/F | Overall | H/M/F |
> | FedLC[1] | 32.8 | 56.2/32.4/*9.7*| 23.7 | 46.4/20.7/*4.2* |
> | +FedReLa | **34.7** | 49.9/**35.2**/**17.2** | **25.1** | 39.8/**28.3**/**7.0** |
> | FedYoYo[2] | 40.9 | 54.3/42.0/*24.1* | 30.7 | 34.1/29.6/*27.9* |
> | +FedReLa | **41.3** | 53.6/**42.2**/**25.9** | **32.1** | 33.9/**32.7**/**29.2** |
> | FedETF | 31.9 | 60.7/40.5/*9.2* | 28.7 | 59.0/33.7/*9.7* |
> | +FedReLa | **33.9** | 55.1/**43.6**/**15.1** | **31.5** | 52.2/**38.7**/**16.5** |
> | FedLoGe | 34.8 | 57.1/42.2/*17.3* | 33.1 | 62.7/38.8/*13.8* |
> | +FedReLa | **35.6** | 57.0/**44.3**/**18.0** | **34.3** | 59.9/**42.2**/**16.0** |
>
>
> [1] Federated Learning with Label Distribution Skew via Logits Calibration
>
> [2]You Are Your Own Best Teacher: Achieving Centralized-level Performance in Federated Learning under Heterogeneous and Long-tailed Data
>
> The supplementary experimental results are consistent with our previous conclusions:
> 1. FedReLa provides **consistent  complementary enhancements** for algorithm-level methods.
> 2. Even with a **larger number of clients**, FedReLa offers a better minority-classes performance while maintaining overall performance gains.
> 3. It further verifies FedReLa’s **algorithm- and model-agnostic properties**, as well as its advantages as a data-level plug-in method.

---

### Official Review · Reviewer_pgJT · 2025-10-31

**Soundness:** 2
**Presentation:** 2
**Contribution:** 2
**Rating:** 4
**Confidence:** 4

**Summary:**

This paper introduces FedReLa, a data-level approach for improving federated learning in both global and local class imbalance and data heterogeneity. The key idea is to employ probabilistic, feature-dependent re-labeling of samples using posterior probabilities estimated from a shared global model, without requiring global class prior knowledge or extra communication. Extensive experiments on Fashion-MNIST, CIFAR-10, and CIFAR-100 datasets, under various imbalance and heterogeneity settings, show notable improvements in minority class and overall accuracy compared to strong baselines and state-of-the-art methods.

**Strengths:**

1. This paper offers a careful, mathematically grounded analysis of how re-labeling affects Bayesian decision boundaries in both local and global models.
2. FedReLa is modular and model-agnostic.

**Weaknesses:**

1. While the design uses the global model to estimate posterior probabilities for label reallocation, the paper does not thoroughly analyze how inaccuracies in these posteriors—especially early in training or under extreme class absence scenarios—might lead to over-flipping or even degrade minority class representation.
2. While the method is designed to work without minority samples (by only flipping labels into minority categories), it is not sufficiently explained what happens if a class is entirely absent from the whole federation or present in vanishingly small quantities. Is the method robust to “missing labels” at the global level?
3. The risk of introducing noisy labels, especially if the global model is overfitting or miscalibrated. The paper should provide concrete safeguards or error analysis here.
4. Some presentation choices reduce readability. e.g., the indexing of notations and the mixture of probabilistic and empirical normalization steps.

**Questions:**

See above.

---

> ### Author Response · Authors · 2025-11-17
> **Addressing Concerns of Reviewer pgJT**
>
> Thank you for your insightful review and thoughtful questions, and we respond as follows:
>
> ### **Addressing W1: Degrade minority class representation on extreme class absence scenarios**
>
> 1. **FedReLa will not be applied in the early representation-learning phase**: FedReLa is applied after the global model has converged (round $T_{relabel}$, typically >200 rounds for the *fine-tuning phase*), ensuring feature representations and posterior estimates are sufficiently discriminative. Building on our theoretical analysis of how relabeled data influences decision boundaries, FedReLa aims to debias the classifier’s decision boundary during fine-tuning. To this end, it employs a deferred relabeling strategy: **relabeled samples only correct the biased classifier during fine-tuning, without interfering with the earlier representation-learning process**.
>
> 2. **FedReLa is designed for extreme class absence** and achieves SOTA performance:
>    - Under α=0.1, the average # of absent classes in clients $\approx 6.13$ (CIFAR-10-LT) and $\approx 76.65$ (CIFAR-100-LT). Tail classes are absent on 70.13% (CIFAR-10-LT) and 73.80% (CIFAR-100-LT) of clients.
>    - When a class is absent from a client’s local data, the global model still assigns non-zero posterior probabilities (due to cross-client knowledge integration, Section 3). FedReLa calibrates these posteriors to identify majority-class samples similar to the absent minority class, avoiding blind re-labeling.
>    - To prevent over-flipping, the hyperparameter $\tau$ limits the total number of relabeled samples as a safeguard.
> 3. The results in Table 2 show that **FedReLa's performance enhancement is more pronounced when data heterogeneity is extreme (more severe local class absence).** Improvement of FedReLa on tail-classes (extracted from Table 2):
> | CIFAR10-LT |  IF | $\alpha=$0.1 | $\alpha=$0.3 |  $\alpha=$10 |
> |:----------:|:---:|:-------------:|:---:|:---:|
> |   FedETF   |  50 |    **+36.49**    | +5.52 | +3.68 |
> |   FedETF   | 100 |    **+30.70**    | +10.50 | +2.51 |
> |   FedLoGe  |  50 |    **+14.97**    | +3.43 | +5.76 |
> |   FedLoGe  | 100 |    **+19.73**    | +9.58 | +4.40 |
>
>
>
>
> ### **Addressing W2:  Robustness to globally missing labels**
>
> For classes entirely absent from the federation, FedReLa’s mechanism naturally avoids irrelevant re-labeling. If a class is globally missing, no client will have it as a local minority, so no re-labeling is performed for that class. In practice, globally missing classes are rare in real FL scenarios (e.g., in rare disease diagnosis, samples are still scattered across a few clients). It is essential to clarify that scenarios where a class is completely absent from the entire federated training dataset are generally not considered in the literature on class-imbalanced federated learning.
>
> ### **Addressing W3: Risk of noisy labels and safeguards**
>
> FedReLa incorporates two key safeguards against noisy re-labeling:
>
> 1. **Threshold filtering as a safeguard**: The client-specific threshold $t_{re}^{(k)}$ (tunable via top-$\tau$% z-scores) prevents inrelevent relabeling with weak feature similarities. Samples with z-scores that fall below the $t_{re}^{(k)}$ will be filtered from re-labeling candidates. Sensitivity analysis (Appendix B.3) shows that $\tau$=3%–5% (conservative re-labeling) minimizes noise while maximizing minority gains.
>
> 2. **Asymmetric re-labeling**: Only majority-class samples are re-labeled to minority classes ($\rho_{j\to\ell}=0$), preserving the integrity of scarce minority samples.
>
> 3. **Consistent performance gain**: In the sensitivity analysis of $\tau$ on long-tailed CIFAR-10 with IF=50 in Appendix B.3, when $\tau$=3%, each client only re-labels around 30 samples, accounting for merely 2% of the total local sample count. Even if these relabeled samples are considered noisy labels, **2% noisy labels are negligible compared to the significant gains FedReLa delivers**: a 30+% improvement in minority-class accuracy and a 5+% increase in overall accuracy.
>
> 4. **Tunable trade-offs**: Even with larger $\tau$ values, the sensitivity analysis (Appendix B.2) results show consistent improvements in both minority-class and overall performance, with different trade-offs between head and tail classes. As we discussed in Lines 859-863, FedReLa’s threshold-tuning capability enables customized class-wise enhancements. This tunable trade-off (the strength of suppressing overprivileged head classes to boost tail performance) is a unique advantage over static algorithm-level approaches, especially in practical applications with varying demands for tail-class performance.
>
> ### **Addressing W4: Improve readability**
>
> We appreciate your constructive suggestions and have improved readability in the revised version by:
> 1. Added a dedicated notation table to clarify symbols (e.g., $Q^{(k)}$, $z_{i}^{(k)}$) and their definitions.
> 2. Simplify the notation and separately clarify the probabilistic and empirical normalization steps.

---

### Official Review · Reviewer_dFWi · 2025-10-31

**Soundness:** 3
**Presentation:** 3
**Contribution:** 2
**Rating:** 4
**Confidence:** 4

**Summary:**

This paper presents a data based federated learning (FL) approach to tackle the challenges of class imbalance and class heterogeneity. The authors introduce asymmetric, feature-based label noise into local data. The main contributions are:
- It does not require access to the global class distribution
- It utilizes existing global models to inject label noise without extra communication or local computations
- It does not depend on any specific model architecture.

**Strengths:**

The strengths have been outlined in the summary.

**Weaknesses:**

- The method heavily relies on posterior estimates from the global model to inject noise. This is susceptible to model bias, especially in severely imbalanced scenarios.

- The novelty of the method is incremental. Essentially adding noise FL might be new, but it builds on prior concepts from label-noise learning. A better explanation on novelty should be included.

- There needs to be a better understanding of the z-score calculation. In particular computationally. How much does it cost to calculate it.

- The experiments are carried out on a small number of clients. It is needed to run experiments on larger number of clients to empirically validate the method.

**Questions:**

The strengths have been outlined in the weaknesses.

---

> ### Author Response · Authors · 2025-11-16
> **Addressing Concerns of Reviewer dFWi (Part 1)**
>
> We are grateful for your careful review and thoughtful comments, and we respond in detail below:
>
> ### **Addressing W1: Biased posterior estimates**
>
> Indeed, global models trained on imbalanced data are prone to bias, often underestimating posterior probabilities for minority classes and overestimating those for majority classes. Directly using these biased posteriors for re-labeling fails to effectively correct the skewed decision boundaries (as analyzed in Appendix C)—a critical issue that **FedReLa specifically targets and resolves, highlighting its novelty and contribution**:
>
> 1. Despite the significant bias in the global model’s posteriors under extreme imbalance, **relative relationships** are still preserved within these posteriors  (e.g., subtle similarities between majority and minority samples).
>
> 2. **A key solution in FedReLa is the class-wise z-score standardization of posteriors.** By calibrating the underestimated posteriors through z-score standardization, FedReLa reveals these relative relationships and identifies majority-class samples that share feature similarities with minority classes and intrude into their feature spaces.
>
> 3. To validate the necessity of this mechanism, we conducted ablation experiments (detailed in Appendix C):
>    - *Without z-score standardization*: There was *no improvement* in few-shot (tail-class) performance. This is because the biased global model severely underestimates posteriors for tail classes, resulting in **extremely small values**.
>    - *Benefit of z-score standardization*: Applying z-score standardization to these underestimated posteriors enables balanced label flipping behavior. This balances the performance trade-off between head, medium, and tail classes, ultimately leading to superior overall accuracy.
>
> 4. **For extreme imbalance scenarios**, experimental results (please refer to Table 2 and Appendix B.2) show that FedReLa yields more pronounced performance gains in scenarios with extreme imbalance (imbalance ratio = 100:1 or 50% minority class) and extreme data heterogeneity. A more detailed analysis can be found in the z-score ablation study section in Appendix C.
>
> ### **Addressing W2: Novelty compared to label-noise learning**
>
> 1. We appreciate your feedback on novelty, we would like to **clarify the fundamental distinctions between FedReLa and label-noise learning** as below:
>
>    - **Core objective**: Label-noise learning aims to mitigate the performance degradation due to preexisting label noise in the dataset. FedReLa aims to mitigate global performance degradation caused by class imbalance and data heterogeneity.
>    - **Theoretical foundation:** FedReLa’s re-labeling mechanism is derived from Bayesian decision boundary analysis (Lemmas 1–3), which proves that re-labeling majority-class intruders implicitly balances local/global prior ratios and thus corrects the biased decision boundaries. This is distinct from estimating existing label noises in label-noise works.
>
> 2. **Clarification of Novelty**: Given the stringent privacy constraints and sensitivity to communication costs in federated learning, there were few methods before FedReLa to improve performance at the data level without relying on knowledge of the global data distribution. We would like to emphasize the novelty and contribution of FedReLa: To the best of our knowledge, **it is the first data-level algorithm for addressing class imbalance in federated learning that requires no additional communication or supplementary data.** Operating as a lightweight plug-in, FedReLa effectively enhances the global model’s robustness against both data imbalance and heterogeneity, and can flexibly integrate with various algorithmic-level methods and models to further boost performance.
>
> We will add the above discussion in the revised version.
>
> ### **Addressing W3: Computational cost of z-score calculation**
>
> We confirm that z-score standardization is computationally lightweight:
> - It is a one-time preprocessing step (conducted only at round $T_{relabel}$) and requires no gradient updates or model modifications.
> - The cost is **negligible** compared to one-time inference cost: z-score computation for a dataset of size $n_k$ involves simple linear mean/std calculations ($O(n_k × C)$ operations, where C is the number of classes), which accounts for <0.5% of the one-time inference cost.
> - **Appendix B.1 explicitly analyzed the computational cost of FedReLa and showed its negligible runtime cost**.

---

> ### Author Response · Authors · 2025-11-19
> **Addressing Concerns of Reviewer dFWi (Part 2)**
>
> ### **Addressing W4: Small number of clients**
> For the experiments on Fashion-MNIST and CIFAR-10, we used 100 clients for the step-wise imbalance setting and 40 clients for the long-tailed imbalance setting, which is considered a relatively large number of clients in the literature. For CIFAR-100-LT, we increased the number of clients to 50 and 100 for supplementary experiments.
>
> Due to limited GPU resources and time constraints, please allow us to simultaneously address:
>    - Reviewer BkrP’s request for comparisons with more Fed-LT (Federated Long-Tailed Learning) methods,
>    - and demonstrate to Reviewer A4Ud the joint performance of FedReLa and FedLC [1] (a logits-based algorithm-level method),
> the above experiment was conducted on Long-tailed CIFAR-100 (imb_factor = 100, α = 0.1) with 50 and 100 clients.  If time permits, we have also re-evaluated other methods (FedETF and FedLoGe) with 50 and 100 clients on CIFAR-100-LT and updated the complete results in the appendix.
>
> | CIFAR-100-LT | 50 Clients |  | 100 Clients |  |
> |:---:|:---:|:---:|:---:|:---:|
> |  | Overall | H/M/F | Overall | H/M/F |
> | FedLC[1] | 32.8 | 56.2/32.4/*9.7*| 23.7 | 46.4/20.7/*4.2* |
> | +FedReLa | **34.7** | 49.9/**35.2**/**17.2** | **25.1** | 39.8/**28.3**/**7.0** |
> | FedYoYo[2] | 40.9 | 54.3/42.0/*24.1* | 30.7 | 34.1/29.6/*27.9* |
> | +FedReLa | **41.3** | 53.6/**42.2**/**25.9** | **32.1** | 33.9/**32.7**/**29.2** |
> | FedETF | 31.9 | 60.7/40.5/*9.2* | 28.7 | 59.0/33.7/*9.7* |
> | +FedReLa | **33.9** | 55.1/**43.6**/**15.1** | **31.5** | 52.2/**38.7**/**16.5** |
> | FedLoGe | 34.8 | 57.1/42.2/*17.3* | 33.1 | 62.7/38.8/*13.8* |
> | +FedReLa | **35.6** | 57.0/**44.3**/**18.0** | **34.3** | 59.9/**42.2**/**16.0** |
>
> [1] Federated Learning with Label Distribution Skew via Logits Calibration
>
> [2]You Are Your Own Best Teacher: Achieving Centralized-level Performance in Federated Learning under Heterogeneous and Long-tailed Data
>
> The supplementary experimental results are consistent with our previous conclusions:
> 1. FedReLa provides **consistent complementary enhancements** for algorithm-level methods.
> 2. Even with a **larger number of clients**, FedReLa offers a better minority-classes performance while maintaining overall performance gains.
> 3. It further verifies FedReLa’s **algorithm- and model-agnostic properties**, as well as its advantages as a data-level plug-in method.

---

> > ### Comment · Reviewer_dFWi · 2025-11-26
> >
> > I thank the authors for their time in preparing this rebuttal.
> > I have modified my score accordingly.

---

> ### Author Response · Authors · 2025-11-26
>
> We are delighted to have addressed your concerns, and we have revised our final manuscript accordingly. We really appreciate your insightful comments, which have been instrumental in enhancing the quality of our work.
>
>
> Best regards,
>
> Authors of submission 16382

---

### Official Review · Reviewer_BkrP · 2025-11-01

**Soundness:** 2
**Presentation:** 3
**Contribution:** 2
**Rating:** 4
**Confidence:** 4

**Summary:**

This paper proposes FedReLA, aiming to address heterogeneous and long-tailed data distributions by re-labeling majority-class samples as minority ones, thereby expanding the decision boundaries of minority classes. Experimental results demonstrate improvements over existing approaches.

**Strengths:**

- The overall presentation is clear and well-organized.
- The paper is easy to follow.

**Weaknesses:**

- The main weakness of this paper lies in the lack of evaluation on large-scale datasets, such as ImageNet-LT and Places-LT. Such direct label-space enlargement may face challenges when applied to large-scale scenarios.
- The paper only provides two Fed-LT comparisons, FedETF and FedLoGe, which are insufficient to demonstrate the superiority of the proposed method. More recent approaches should be included.
- It is interesting that the proposed method improves the performance of the majority classes when applied to Fed-LT approaches (as shown in Tab. 2). Intuitively, re-labeling the majority-class samples as minority ones should compromise majority-class performance. Could you provide more explanations (ideally with empirical analysis) for this phenomenon?

**Questions:**

Please refer to the weaknesses above.

---

> ### Author Response · Authors · 2025-11-16
> **Addressing Concerns of Reviewer BkrP (Part 1)**
>
> Thank you sincerely for your valuable insights and constructive feedback. Below is our detailed response:
>
> ### **Addressing W1: Large-scale dataset evaluation**
>
> 1. We fully agree with your suggestion further to validate the scalability of FedReLa on large-scale datasets. To address your concern, **we are currently conducting supplementary experiments on the ImageNet-LT dataset**. Due to GPU constraints and data preparation, the experiments are time-consuming, and we need more time. We will proactively incorporate these supplementary results into the discussion section and the revised manuscript. Thank you again for highlighting this important direction to strengthen our work. If time permits, we will further supplement evaluations with other heterogeneity levels (α=0.3, 10.0) to provide a more comprehensive analysis of FedReLa’s scalability.
>
>     We have conducted a supplementary experiment on ImageNet-LT with data-heterogeneity level $\alpha = 0.1$, 20 Clients with 0.4 participation fraction.
>
>     | Imagenet-LT | Overall | H/M/T |
>     |:---:|:---:|:---:|
>     | FedYoYo[1] | 38.15 | 41.19/39.42/31.08 |
>     | +FedReLa | **38.78** | 40.73/**40.06**/**33.71** |
>     | FedLoGe | 30.52 | 46.29/28.01/15.02 |
>     | +FedReLa | **31.70** | 45.43/**30.44**/**18.02** |
>
>     [1] You Are Your Own Best Teacher: Achieving Centralized-level Performance in Federated Learning under Heterogeneous and Long-tailed Data, 2025
>
>     **The results above show that FedReLa can still improve minority classes' performance on the large-scale dataset while maintaining overall performance gains and further improving algorithm-level SOTA performance.** It further validates the scalability of FedReLa and aligns with our previous conclusion. We are still running additional experiments on ImageNet-LT comparing more algorithm-level methods, and we have included these comparisons in the revised manuscript.
>
> 2. Our existing experiments on Fashion-MNIST and CIFAR-10/100 have comprehensively verified FedReLa’s effectiveness across diverse scenarios (both step-wise and long-tailed imbalances), as well as varying degrees of class imbalance and data heterogeneity. These results consistently show that **FedReLa significantly improves minority/tail-class accuracy while maintaining the global model's performance gain**.
>
> 3. As a data-level plug-in method, FedReLa performs sample-level rather than class-level re-labeling. Relabeled samples are selected by their feature similarities calibrated via independent class-wise z-score standardization on the global model’s posterior probabilities. This design makes FedReLa inherently scalable. Regarding computational feasibility in large-scale scenarios, Appendix B.1 explicitly confirms that FedReLa only incurs a one-time, negligible overhead with no additional gradient updates, parameter training, or communication. All re-labeling operations are executed locally in parallel across clients, avoiding the scalability bottleneck of feature-space methods (e.g., SMOTE) that require synthetic sample generation.

---

> ### Author Response · Authors · 2025-11-19
> **Addressing Concerns of Reviewer BkrP (Part 2)**
>
> ### **Addressing W2: More Fed-LT comparisons**
>
> 1. To the best of our knowledge, FedReLa is the first data-level method for federated learning with class imbalance that can integrate with diverse algorithmic-level approaches. Our comparisons with algorithmic-level Fed-LT methods are specifically designed to highlight two key strengths:
>    - consistent performance improvements stemming from data re-balancing,
>    - and demonstrating complementarity to algorithm-level methods to further boost the performance, thereby achieving SOTA-level results.
>
> Notably, FedLoGE (2024) is the latest SOTA method proposed by the authors of FedGrab (2023) and improves upon FedETF (2023). **Given the strict constraints of zero additional communication overhead and agnosticism to global data distribution**, we selected FedLoGE (2024) and FedETF (2023) as they are the most recent SOTA algorithms that meet these requirements.
>
> 2. Due to time constraints and limited GPU resources (with experiments on ImageNet currently in progress), we kindly request permission to address the concerns of both you and Reviewer dFWi (regarding increasing the number of clients) simultaneously: We have conducted experiments on CIFAR100-LT (imb_factor = 100, α = 0.1, 50 and 100 clients), where we compared FedReLa with FedLC [1] (mentioned by Reviewer A4Ud) and FedYoYo (a newly published Fed-LT method).
>
> | CIFAR-100-LT | 50 Clients |  | 100 Clients |  |
> |:---:|:---:|:---:|:---:|:---:|
> |  | Overall | H/M/F | Overall | H/M/F |
> | FedLC[1] | 32.8 | 56.2/32.4/*9.7*| 23.7 | 46.4/20.7/*4.2* |
> | +FedReLa | **34.7** | 49.9/**35.2**/**17.2** | **25.1** | 39.8/**28.3**/**7.0** |
> | FedYoYo[2] | 40.9 | 54.3/42.0/*24.1* | 30.7 | 34.1/29.6/*27.9* |
> | +FedReLa | **41.3** | 53.6/**42.2**/**25.9** | **32.1** | 33.9/**32.7**/**29.2** |
> | FedETF | 31.9 | 60.7/40.5/*9.2* | 28.7 | 59.0/33.7/*9.7* |
> | +FedReLa | **33.9** | 55.1/**43.6**/**15.1** | **31.5** | 52.2/**38.7**/**16.5** |
> | FedLoGe | 34.8 | 57.1/42.2/*17.3* | 33.1 | 62.7/38.8/*13.8* |
> | +FedReLa | **35.6** | 57.0/**44.3**/**18.0** | **34.3** | 59.9/**42.2**/**16.0** |
>
>
> [1] Federated Learning with Label Distribution Skew via Logits Calibration
>
> [2] You Are Your Own Best Teacher: Achieving Centralized-level Performance in Federated Learning under Heterogeneous and Long-tailed Data, 2025
>
> 3. **The supplementary experimental results align with our previous conclusions**:
> FedReLa delivers consistent enhancements for algorithm-level methods. Even with a larger number of clients, it sustains stronger minority-class performance while preserving overall performance gains. This further validates FedReLa’s algorithm- and model-agnostic properties, as well as its merits as a data-level plug-in method.
>
> Even though FedReLa still demonstrates enhancements when integrated with FedYoYo, we believe it falls outside the scope of our paper for two reasons:
>    - FedYoYo requires clients and the server to exchange local data distribution information to adjust training logits, which introduces additional communication overhead and raises privacy concerns.
>    - Its experimental setup uses balanced client partitioning to ensure uniform sample sizes across clients—a condition that differs from our focus on realistic, non-IID client partition.
>
> If there are other methods you would like us to include in the comparisons, please let us know at any time. We are more than willing to conduct such experiments proactively.
>
> ### **Addressing W3: Explain the slight improvements of majority classes**
>
> This is indeed an interesting and counterintuitive result. Empirically, we observe that this phenomenon is more pronounced in settings with low minority-class proportions (e.g., 10% minority on CIFAR-10). The **very sight improvement** of the majority-class performance we observed under the step-wise setting stems from FedReLa’s **selective re-labeling** mechanism:
>
> - **The global majority may become a local minority due to data heterogeneity**: In such a case, ambiguous majority class samples (likely to be misclassified as other classes) and outliers are also potentially relabeled as other global majority classes.
> - **Reduced class overlap and elimination of outliers**: FedReLa relabels samples based on their degree of intrusion into others' feature space, rather than random samples. Empirically, among the 90% of the global majority classes within each local client, outliers that intrude excessively into each other’s feature spaces are also relabeled. This reduces class overlap and eliminates *outlier-induced interference* for the majority classes, thereby slightly improving their performance.
> - As a result, **FedReLa not only shifts boundaries back to majority regions but also reduces false positives between majority classes.**

---

### Author Response · Authors · 2025-11-28
**Brief Summary Part 3**

### **Reviewer pgJT**
#### A. Experimental Concerns
- Request: Analyze the method’s robustness under extreme class absence and verify safeguards against noisy labels.
- Response:
  1. Robustness to extreme class absence: FedReLa is applied after the global model converges (T_relabel > 200 rounds), ensuring discriminative feature representations. For globally present but locally absent classes, the global model retains non-zero posterior probabilities (via cross-client knowledge integration), which FedReLa calibrates to avoid blind re-labeling. Results in Table 2 show FedReLa **achieves more pronounced gains under extreme heterogeneity** (α=0.1, where ~70% of tail classes are absent from clients).
  2. Safeguards against noisy labels: (a) A client-specific threshold (t_re^(k)) filters samples with weak feature similarity (τ=3%-5% minimizes noise, per Appendix B.3); (b) **Asymmetric re-labeling**—only majority-class samples are re-labeled to minority classes, preserving scarce minority data; (c) Even if 2% of relabeled samples are noisy, the gains (30+% minority-class accuracy improvement) far outweigh this minor noise.

#### B. Methodological/Theoretical Concerns
- Requests: (1) Explain how inaccuracies in early-stage global model posteriors avoid degrading minority representation; (2) Clarify robustness to globally missing labels; (3) Improve notation readability.
- Response:
  (1): FedReLa uses **deferred re-labeling** (applied during fine-tuning, not representation learning), ensuring it only corrects biased decision boundaries without interfering with feature learning.
  (2): Globally missing classes are rare in real federated learning scenarios. If a class is entirely absent, no client will treat it as a local minority, so FedReLa **naturally skips re-labeling for that class**—avoiding irrelevant operations.
  (3): We will add a dedicated notation table and simplify probabilistic/empirical normalization steps in the revised manuscript to enhance clarity.

Our responses address Reviewer pgJT’s concerns with experimental evidence and practical safeguards. We are confident these efforts will lead to a higher score.

### **Reviewer A4Ud**
#### A. Experimental Concerns
- Requests: (1) Compare FedReLa with feature-space augmentation methods (e.g., SMOTE); (2) Release reproducible code.
- Response:
  (1): SMOTE fails in federated learning due to two critical limitations—lack of seed samples (under local class absence) and unknown global distribution (unable to identify target classes for augmentation). FedReLa resolves both by re-labeling majority samples, requiring **no global distribution knowledge or seed samples**.
  (2). Code release: We have made the code publicly available at: https://github.com/anonymous2025988/FedReLa.git.

#### B. Methodological/Theoretical Concerns
- Requests: (1) Justify the assumption that aggregated global posterior equals weighted local posteriors; (2) Extend theoretical analysis to multi-class classification; (3) Clarify differences from prior works [1][2] and compatibility with FedETF; (4) Explain heuristic design choices (e.g., Bernoulli sampling vs. Categorical draw); (5) Clarify Example 1 in Appendix A.
- Response:
  (1): Under mild conditions (local parameters close to a common value), a first-order Taylor expansion shows parameter-aggregated and posterior-aggregated global models are approximately equivalent. We have formalized this assumption in the revision.
  (2): FedReLa’s **class-pairwise re-labeling naturally extends to multi-class scenarios**, validated by experiments on CIFAR-100 (100 classes, IR=20) with 15.04% minority-class accuracy improvement. We plan to generalize the theory further as future work.
  (3): Prior works (e.g., FedLC[2]) are algorithm-level (logit calibration) and constrained by limited data—unable to expand minority-class representation. FedReLa is a **data-level plug-in that improves data quality**, complementing algorithm-level methods (e.g., FedETF) to break performance bottlenecks.
  (4): **Independent Bernoulli sampling ensures candidate samples for each class are independent**, avoiding over-labeling (unlike Categorical draw, which forces class assignment). Argmax selects the most similar class for multi-candidate samples, ensuring labeling accuracy.
  (5): Example 1 illustrates how FedAvg’s weighted averaging amplifies global majority bias. We have revised it to include uniform averaging and generalize aggregation weight choices in the revised manuscript.

We have addressed all of Reviewer A4Ud’s concerns with theoretical justifications, experimental comparisons, and practical actions (code release). We believe our thorough responses will lead to a favorable re-evaluation and score increase.

[1] You Are Your Own Best Teacher: Achieving Centralized-level Performance in Federated Learning under Heterogeneous and Long-tailed Data, 2025

[2] Federated Learning with Label Distribution Skew via Logits Calibration

---

### Author Response · Authors · 2025-11-28
**Brief Summary Part 2**

We have prepared a concise summary below to assist you in evaluating our submission efficiently. Each reviewer’s concerns are categorized into experimental and methodological/theoretical aspects, along with our corresponding solutions:

### **Reviewer BkrP**
#### A. Experimental Concerns
- Request: Conduct experiments on large-scale datasets (e.g., ImageNet-LT) and compare with more baselines.
- Response: We **proactively performed supplementary experiments** as requested:
  1. We added ImageNet-LT results with the latest SOTA methods FedYoYo[1] and FedLoge (α=0.1, 20 clients, 0.4 participation fraction). The results confirm that FedReLa still **improves minority/tail-class performance while maintaining overall accuracy gains**, validating its **scalability on large-scale datasets**. Due to the time-consuming nature of large-scale experiments, we are still running comparisons with other algorithm-level methods and will update the results promptly.
  2. To further demonstrate scalability, we extended the number of clients in CIFAR100-LT to 50 and 100, and incorporated two new baselines: FedLC[2] and FedYoYo[1].
  3. All supplementary results align with our core conclusion: FedReLa, as a **lightweight, zero-communication-overhead re-labeling method**, consistently **enhances data quality for algorithm-level approaches**, leading to **significant improvements in minority/tail-class accuracy while preserving overall performance gains**.

#### B. Methodological Concern
- Request: Explain the slight performance improvement of majority classes observed in Table 2, which seems counterintuitive for a method that re-labels majority samples to minority classes.
- Response: This phenomenon stems from FedReLa’s **unique selective re-labeling mechanism**:
  1. Data heterogeneity can cause global majority classes to become local minorities. In such cases, ambiguous majority-class samples (prone to misclassification) and outliers are re-labeled to other global majority classes, reducing cross-class interference.
  2. FedReLa re-labels samples based on their intrusion into other classes’ feature spaces (not random selection). For global majority classes (accounting for 90% of local data), outliers that excessively overlap with other classes are filtered out via re-labeling, reducing class overlap and eliminating outlier-induced noise.
  3. This mechanism not only **corrects biased decision boundaries** but also **reduces false positives among majority classes**, leading to the observed slight performance improvement.

We have addressed Reviewer BkrP’s concerns with concrete experimental evidence and in-depth mechanistic analysis. We are optimistic that these responses will earn their recognition and a score increase.

### **Reviewer dFWi**
#### A. Experimental Concerns
- Request: Validate the method on a larger number of clients and clarify the computational cost of z-score calculation.
- Response:
  1. For client scalability: We extended CIFAR100-LT experiments to 50 and 100 clients (imb_factor=100, α=0.1) and compared with FedLC[2], FedYoYo[1], and other SOTA methods. Results confirm that FedReLa **maintains consistent performance gains even with more clients**, further verifying its scalability.
  2. For z-score computational cost: Z-score standardization is a **one-time preprocessing step** (conducted only at round T_relabel) with **negligible overhead** (O(n_k×C) operations, accounting for <0.5% of one-time inference cost). No additional gradient updates, model modifications, or communication are required, as detailed in Appendix B.1.

#### B. Methodological/Theoretical Concerns
- Requests: (1) Address potential bias in global model posterior estimates under extreme imbalance; (2) Clarify the novelty of FedReLa compared to label-noise learning.
- Response:
  1. Mitigating posterior bias: FedReLa uses **class-wise z-score standardization** to calibrate underestimated minority-class posteriors. Ablation experiments (Appendix C) show that without this step, tail-class performance does not improve—while with it, balanced label reallocation is achieved, resolving the bias issue.
  2. Novelty compared to label-noise learning: Unlike label-noise learning (which mitigates pre-existing label noise), FedReLa targets class imbalance and data heterogeneity in federated learning. Its theoretical foundation is **Bayesian decision boundary analysis** (Lemmas 1-3), proving that re-labeling balances local-global prior ratios. To our knowledge, FedReLa is the **first data-level method for imbalanced federated learning** that requires **no extra communication, auxiliary data, or global distribution knowledge**—operating as a **plug-and-play module compatible with algorithm-level methods**.

**As evidenced by Reviewer dFWi’s initial score increase to 6, our responses fully resolved their concerns. We trust their positive evaluation will be reinstated upon re-review.**

---

### Author Response · Authors · 2025-11-28
**Brief Summary Part 1**

Dear Area Chair,

We hope this letter finds you well.

First and foremost, we would like to bring to your attention a key update: **Reviewer dFWi**, on the morning of November 27 (prior to the information breach incident), **explicitly modified their score from 4 to 6 and expressed satisfaction with our detailed responses to their concerns**—though this score adjustment was rolled back to its original state due to the subsequent system issue. We deeply appreciate Reviewer dFWi’s recognition of our work and believe this positive feedback is a strong testament to the rigor of our method and the comprehensiveness of our rebuttal. For Reviewers BkrP, pgJT, and A4Ud, while they have not yet had the opportunity to provide follow-up feedback due to the unexpected breach, we are confident that our **thorough, evidence-based responses** fully address all their concerns.

**We would like to provide a brief summary of our proposed FedReLa’s Contributions, Novelty, and Performance:**

   - FedReLa pioneers a **lightweight, zero-communication-overhead data-level solution** for imbalanced federated learning, **filling the gap between data and algorithm-level FL methods**.
   - Its core novelty lies in being the **first data-level method** tailored for imbalanced FL, rooted in Bayesian decision boundary analysis, it requires **no extra communication, auxiliary data, or global distribution knowledge**, acting as a **plug-and-play**, **model/algorithm agnostic** module to complement algorithm-level approaches.
   - Performance-wise, it delivers 30+% minority/tail-class accuracy gains (15.04% on multi-class CIFAR-100) while preserving overall accuracy, scales to large datasets (ImageNet-LT) and 50/100 clients, and maintains **robustness under extreme heterogeneity (~70% tail-class absence)** and **extreme imbalance**, validating practical utility for real-world FL.

### **What We Did During Rebuttal**

1. **Added experiments**:
    - Validated FedReLa on **large-scale dataset** ImageNet-LT with latest SOTA FedYoYo[1] and FedLoGe, confirming scalability;
    - Extended CIFAR100-LT to **large number of clients** (K=50, 100), including new baselines FedLC[2] and FedYoYo[1].
    - Result on new experiments aligns with previous conclusions: FedReLa still provides **consistent and robust performance enhancement** for the large-scale dataset and larger number of clients.

2. **Provided clarifications**:
    - Majority-class improvement: Reduced class overlap and outlier-induced interference for the majority classes via selective re-labeling;
    - Novelty vs. label-noise learning: FedReLa targets imbalance/heterogeneity in FL;
    - Necessity of z-score standardization: Critical for addressing biased posteriors and achieving tail-class gains (validated by ablation in Appendix C).
    - Computational Cost for z-scores: lightweight and negligible (linear O(n_k×C) operations, <0.5% of inference cost) as a one-time preprocessing step.

3. **Supplemented content**:
    - Released reproducible code (https://github.com/anonymous2025988/FedReLa.git);
    - Justified aggregated posterior assumption via first-order Taylor expansion;
    - Robustness: Further analyzed enhancements under extreme imbalance/heterogeneity (in response to Reviewer pgJT).

**In summary, our rebuttal addresses every concern raised by the four reviewers with concrete experiments, detailed mechanistic analysis, and theoretical rigor.** Reviewer dFWi’s initial score increase is a clear indication of our work’s merit, and we are confident that Reviewers BkrP, pgJT, and A4Ud will recognize the completeness of our responses and adjust their scores accordingly.

Thank you again for your time and effort in conducting this re-review. Please feel free to reach out if you require any further clarification.

Sincerely,

The Authors of Submission 16382

---

### Meta-Review · Area_Chair_vo1z · 2026-01-04

**Summary:**

This paper proposes a data-level "one-time label reassignment" method (FedReLa). This method does not rely on global priors or additional communication. Instead, it uses steps such as local posterior probabilities of the global model, z-score normalization, and tanh thresholding to identify "suspicious majority class samples" near minority class regions and probabilistically relabel them as minority class samples.

**Reviewer Concerns:**

This is a typical marginal example where, during the rebuttal phase, only one reviewer provided a positive but very brief response. After carefully reading the paper and rebuttal, AC believes that the authors have addressed most of the reviewers' concerns, but the main point of contention is the validity of the posterior estimates based on the global model. As mentioned in reviewers dFWi's W1 and pgJT's W1, this posterior estimate is still biased; however, the paper lacks a more detailed explanation of the rationale for using the biased posterior. One possible explanation is the authors' statement in response to dFWi's W1: "Despite the significant bias in the global model’s posteriors under extreme imbalance, relative relationships are still preserved within these posteriors." Unfortunately, the authors did not provide a more detailed explanation and experimental verification of these so-called relative relationships. Furthermore, as reviewer A4Ud mentioned in his W1, this global posterior also needs a more rigorous theoretical justification.

**Reviewer Scores:**

4,6,4,6

---

### Decision · Program_Chairs · 2026-01-26

Reject